# SPURIOUS PRIVACY LEAKAGE IN NEURAL NETWORKS

## ABSTRACT

Neural networks are vulnerable to privacy attacks aimed at stealing sensitive data. When trained on real-world datasets, these models can also inherit latent biases, which may further increase privacy risks. In this work, we investigate the impact of spurious correlation bias on privacy vulnerability, identifying several key challenges. We introduce *spurious privacy leakage*, a phenomenon where spurious groups can be more vulnerable to privacy attacks than non-spurious groups, and demonstrate how this leakage is connected to task complexity. Furthermore, while robust training methods can mitigate the performance disparity across groups, they fail to reduce privacy vulnerability, and even differential privacy is ineffective in protecting the most vulnerable spurious group in practice. Finally, we compare model architectures in terms of both performance and privacy, revisiting prior research with novel insights.

## 1 INTRODUCTION

Machine learning models are applied across various domains such as face recognition, medical prognosis, or personalized advertisement. All these domains require models to be trained on user-sensitive data that can be of interest to attackers (Shokri et al., 2017; Liu et al., 2021a; Mireshghallah et al., 2020; Yeom et al., 2018). Such a potential data leak breaches the required property of *privacy-preserving* data management, which aims to protect data confidentiality. In addition to privacy concerns, models trained on real-world and high-dimensional data can develop biases towards specific groups, a subset of the dataset sharing a common characteristic (e.g. gender, ethnicity, or geographic location). These biases can cause models to output unfair and inaccurate predictions in deployment, which may also increase privacy vulnerabilities, all due to the learning of misleading features (Sagawa et al., 2019; Geirhos et al., 2020; Shah et al., 2020). Therefore, models deployed in sensitive applications should satisfy multiple constraints, such as ensuring fair performance across subpopulations while also guaranteeing the protection of sensitive data.

In this work, we focus on the *spurious correlation* bias, a statistical relationship between two variables that appears to be causal but is either caused by a third confounding variable or random chance. While spurious correlations have been widely studied in machine learning, current spurious robust methods primarily address group performance disparity, overlooking other concerns like privacy (Izmailov et al., 2022; Yang et al., 2023). On the other hand, privacy research typically evaluates methods using balanced datasets such as Adult, CIFAR, or ImageNet (Hu et al., 2022). We fill these gaps by investigating the privacy of neural networks trained on biased real-world datasets using *membership inference attacks* (MIA), a family of privacy attacks commonly used for their simplicity and versatility (Murakonda & Shokri, 2020; Carlini et al., 2021). In the presence of spurious correlation bias, we find a phenomenon we term *spurious privacy leakage*, where certain groups with spurious correlation are significantly more vulnerable to MIA than others, raising additional security challenges. For example, privacy auditing may naively conclude that a model satisfies the requirements using aggregated metrics over the whole dataset. However, we find that spurious correlation can cause one group to be significantly more vulnerable than others, violating the requirements for that group. Studying privacy disparity is important to precisely understand the risks our models carry, to encourage the research of more robust defenses, and to improve the auditing process. Furthermore, GDPR enforces equal treatment (European Union, 2016), making the oversight of disparities a potential compliance risk.

We perform a set of experiments to explore the broader implications of *spurious privacy leakage*. While previous works suggested that improving the generalization across groups can mitigate privacy disparity (Kulynych et al., 2022), to the best of our knowledge, there is no evidence to support

this claim. To address this, we evaluate the privacy vulnerabilities of models using spurious robust training, which are designed to improve the worst group performance (Sagawa et al., 2019; Kirichenko et al., 2022; Izmailov et al., 2022) while their privacy side effects are unknown. Additionally, we assess the effectiveness of differential privacy as a defense for models trained on spurious correlated datasets. Finally, while prior works mostly focus on the ResNet-like architecture for privacy analysis (Carlini et al., 2022; Liu et al., 2022a), we comprehensively and fairly compare the performance and privacy of eight model architectures, including state-of-the-art convolutional and transformer-based, supervised and self-supervised pretrained architectures.

**Contributions.** We investigate how learning with natural spurious correlations affects privacy vulnerability. Using real-world datasets, we reveal *spurious privacy leakage*, a phenomenon where the groups affected by spurious correlations can be up to 100 times more vulnerable to membership inference attacks than non-spurious groups (Section 3.1), and we demonstrate how this leakage emerges as the task complexity of the dataset simplifies (Section 3.2). Furthermore, we are the first to observe that robust training methods can reduce group performance disparity but not the privacy disparity (Section 4.1). We present the practical limitations of using differential privacy with spurious correlations, showing that despite the drop in utility, there is no meaningful gain in privacy protection for spurious groups (Section 4.2). Finally, we show that the choice of model architecture significantly impacts both performance and privacy disparity (Section 5). Our code is available at `https://anonymous.4open.science/r/spurious-mia-6676`.

## 2 BACKGROUND

We provide a concise introduction of concepts needed to follow the rest of the work such as machine learning, membership inference attacks, and spurious correlation.

**Neural networks** represent functions $f_{\boldsymbol{\theta}} \colon \mathcal{X} \to \mathcal{Y}$ that map the input data $\boldsymbol{x} \in \mathcal{X}$ to a label $\boldsymbol{y} \in \mathcal{Y}$. The dataset $\mathcal{D} = \{(\boldsymbol{x}_i, \boldsymbol{y}_i)\}$ is a set of labeled pairs used for estimating the model parameters. The neural network is parametrized by $\boldsymbol{\theta} \in \mathbb{R}^n$ and it is updated using a first-order optimization method (e.g. stochastic gradient descent) to minimize a loss function $\ell \colon \mathcal{Y} \times \mathcal{Y} \to \mathbb{R}$. We focus on the classification setting where the cross-entropy loss is commonly used. Formally, the objective is the *empirical risk minimization* (ERM) (Vapnik, 1991) using the cross-entropy loss:

$$\hat{\boldsymbol{\theta}}_{\mathrm{ERM}} = arg \min_{\boldsymbol{\theta}} \mathbb{E}_{(\boldsymbol{x},\boldsymbol{y}) \in \mathcal{D}}(\ell(\boldsymbol{y}, f_{\boldsymbol{\theta}}(\boldsymbol{x})) \qquad \ell_{\mathrm{CE}}(\boldsymbol{y}, \boldsymbol{p}) = -\sum_{i=1}^{c} y_i \log(\boldsymbol{p})$$

where $c$ is a scalar representing the number of target classes, $\boldsymbol{y}$ is the one-hot label encoding vector, and $\boldsymbol{p}$ is the model's output as a probability vector.

**Membership inference attacks** (MIA) aim to determine whether a specific input data was used during the model training. MIA is usually used to audit a model's privacy level thanks to its simplicity (Murakonda & Shokri, 2020) and versatility for creating a more complex attack (Carlini et al., 2021). The membership inference problem can be defined as learning a function $\mathcal{A} \colon \mathcal{X} \to [0, 1]$, where $\mathcal{A}$ is the attacker model that takes input $\boldsymbol{x} \in \mathcal{X}$ and outputs 1 if $\boldsymbol{x}$ was used during the model training. Generally, the attacker assumes to either have white-box (Nasr et al., 2019) or black-box (Shokri et al., 2017) access to the target model, depending on the amount of information the target reveals. Black-box access is when the only target information accessible is the output probability vector $\boldsymbol{p}$, while white-box access relaxes the condition by providing additional information such as the type of architecture or the training algorithms.

Shokri et al. (2017) introduced the first MIA for neural networks assuming a black-box access, where several *shadow* models are trained to mimic the behavior of the *target* model. More advanced attacks have been developed based on the idea of shadow models (Yeom et al., 2018; Liu et al., 2022a; Carlini et al., 2022; Ye et al., 2022; Sablayrolles et al., 2019; Watson et al., 2021; Long et al., 2020). In this work, we focus on the state-of-the-art LiRA method (Carlini et al., 2022). Given an input $\boldsymbol{x}$, LiRA predicts its membership by training $N$ shadow models, each on a different subset of the dataset. Half of the models are named INs and contain $\boldsymbol{x}$ and the other half named OUTs do not. Each shadow model IN outputs a confidence score $\phi(\boldsymbol{p}_{\mathrm{shadow}})$ which is used to estimate the parameters of a Gaussian $\mathcal{N}(\mu_{\mathrm{in}}, \sigma_{\mathrm{in}})$, and in the same way, OUTs are used to estimate $\mathcal{N}(\mu_{\mathrm{out}}, \sigma_{\mathrm{out}})$. Finally, the result of the attack is defined as a likelihood-ratio test:

$$\Lambda = \frac{\Pr(\phi(\boldsymbol{p}_{\text{target}}) \mid \mathcal{N}(\mu_{\text{in}}, \sigma_{\text{in}}))}{\Pr(\phi(\boldsymbol{p}_{\text{target}}) \mid \mathcal{N}(\mu_{\text{out}}, \sigma_{\text{out}}))} \qquad \phi(\boldsymbol{p}) = log(\frac{\boldsymbol{p}}{1 - \boldsymbol{p}})$$

where $\phi(\boldsymbol{p}_{\text{target}})$ is the confidence score obtained by querying the target model with $\boldsymbol{x}$. The score $\Lambda$ is used by the attacker to determine how likely it is that the given $\boldsymbol{x}$ is a member.

**Spurious correlation** is a statistical relationship between two variables $X$ and $Y$ that first appears to be causal but in reality is either caused by a third confounding (e.g. spurious) variable $Z$ or due to random chance. This relationship is in contrast with causality, where the change of the variable $X$ leads to a direct and predictable outcome of $Y$ while ruling out the presence of any confounding factors $Z$. For a given dataset with spurious correlation, a feature $\boldsymbol{z}$ is called spurious if it is correlated with the target label $\boldsymbol{y}$ in the training data but not in the test data. For example, in a binary bird classification dataset where waterbirds mainly appear on a water background, a biased model can exploit the background spurious feature instead of the bird invariant feature, leading to a wrong prediction when the input is a waterbird on a land background (Sagawa et al., 2019). Ideally, we would like to suppress the bias coming from the spurious features, which can be expressed as $\Pr(\boldsymbol{y} \mid \boldsymbol{x}) = \Pr(\boldsymbol{y} \mid \boldsymbol{x}_{\text{inv}}, \boldsymbol{z}) = \Pr(\boldsymbol{y} \mid \boldsymbol{x}_{\text{inv}})$ where we decomposed the input $\boldsymbol{x}$ as a combination of invariant features $\boldsymbol{x}_{\text{inv}}$ and spurious features $\boldsymbol{z}$.

Sagawa et al. (2019) proposed the group *distributionally robust optimization* (DRO) to mitigate spurious features. DRO minimizes the worst-group loss, differing from ERM which minimizes the average loss. Formally, the objective function of DRO is defined as:

$$\hat{\boldsymbol{\theta}}_{\text{DRO}} = arg \min_{\boldsymbol{\theta}} \max_{\boldsymbol{g} \in \mathcal{G}} \mathbb{E}_{(\boldsymbol{x}, \boldsymbol{y}, \boldsymbol{g}) \in \mathcal{D}}[\ell(\boldsymbol{y}, f_{\boldsymbol{\theta}}(\boldsymbol{x}))]$$

where the dataset is divided into $\boldsymbol{g}$ groups. The new dataset is $\mathcal{D} = \{(\boldsymbol{x}_i, \boldsymbol{y}_i, \boldsymbol{g}_i)\}$ where $\boldsymbol{g} \in \mathcal{G}$ is a discrete-valued label (e.g. all the combinations of birds and backgrounds or geographical area information (Koh et al., 2021)). DRO is considered an oracle method due to its explicit use of the group information for the training (Liu et al., 2021b). Additional methods in the literature suppress the spurious features by learning and assigning a different weight per sample (Liu et al., 2021b; Nam et al., 2020), by retraining the classifier head at the end of the training (Kirichenko et al., 2022; Izmailov et al., 2022; Kang et al., 2019), by group sampling (Yang et al., 2024; Idrissi et al., 2022), or using contrastive methods (Zhang et al., 2022). In particular, Kirichenko et al. (2022) developed *deep feature reweighting* (DFR) to mitigate spurious correlation by simply retraining the last-linear layer of an ERM trained model using a group-balanced dataset.

## 3  SPURIOUS CORRELATION AND PRIVACY RISKS

We demonstrate the differences in privacy leakage between spurious and non-spurious correlated groups. Our results show that auditing the privacy level on the whole dataset is misleading in the presence of spurious correlations (Carlini et al., 2022; Feldman & Zhang, 2020) where the spurious groups can have significantly higher privacy leakage.

### 3.1  SPURIOUS PRIVACY LEAKAGE

Spurious correlations are characterized by the presence of spurious features. Assuming we have the labels of the spurious features, learning with spurious correlation is equivalent to learning with an imbalanced dataset. We refer to spurious groups as the minority groups with the worst performance (e.g. worst-group accuracy) compared to the majority groups.

*Experiment setup.* We choose the datasets that are commonly used by the spurious correlation community (see Table 2 from Yang et al. (2023)): Waterbirds (Sagawa et al., 2019), CelebA (Liu et al., 2014), FMoW (Koh et al., 2021), and MultiNLI (Williams et al., 2017). These datasets contain real-world spurious correlations, diverse modalities, and different target complexity. Moreover, to the best of our knowledge, we are the first to study MIA attacks on subgroups of these datasets. Appendix A provides further details for each dataset. We use the pretrained ResNet50 (He et al., 2016) on ImageNet1k from the timm[1] library and finetune using random crop and horizontal flip. Our

---

[1]https://github.com/huggingface/pytorch-image-models

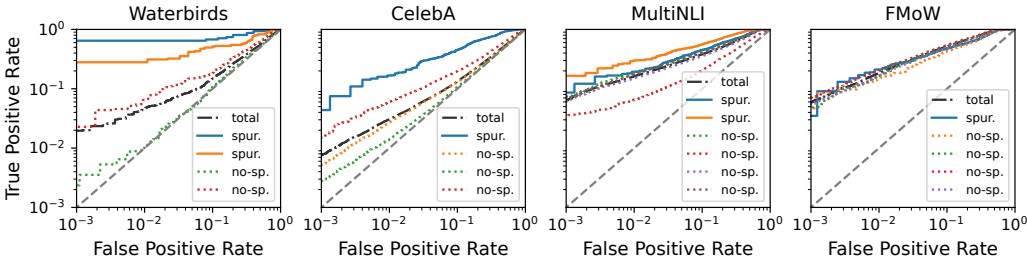

Figure 1: Attack success rate divided per group on Waterbirds, CelebA, MultiNLI, and FMoW respectively. Across the datasets, there is a spurious group (solid lines) with higher privacy leakage compared to non-spurious groups under the LiRA privacy attack. However, all the groups in FMoW have similar levels of leakage, which we investigate in Section 3.2.

setting differs from the standard settings (i.e. training from random initialization) by initializing our models using pretrained weights on public data. We perform hyperparameter optimization for each dataset using a grid search over learning rate (lr), weight decay (wd), and epochs. The grid search and its best configuration are in Appendix B. We report the training and test accuracy to evaluate the performance and their difference to quantify the overfitting level. We do the same for the worst-group accuracy (WGA), which is a commonly used proxy metric to measure the mitigation success of spurious features (Sagawa et al., 2019). For privacy evaluation, we follow the guidelines from Carlini et al. (2022). We train non-overfit models and report the full log-scale ROC curves, the true positive rate (TPR) at a low false positive rate (FPR) region, and also the AUROC curve for completeness. We train 32 shadow models for Waterbirds/CelebA and 16 for FMoW/MultiNLI.

Across all the spurious correlated datasets, there is always a group performance disparity (see Table 5). For example, in the Waterbirds dataset, ERM training has a test average accuracy of 81.08%, while if we account for only the spurious group, it is 34.41%. Beyond group performance disparity, we show that spurious correlations also cause privacy issues, leading to the phenomenon of *spurious privacy leakage*. Using the state-of-the-art MIA method LiRA (Carlini et al., 2022), we analyze the privacy leakage for each group of the four real-world datasets commonly used within the spurious correlation community. For each dataset, we train the shadow models using 50% of the sampled training data as described by the LiRA algorithm. We ensure that the sampled subset maintains a similar group proportion as the original dataset by first sampling per group, and then combining all the sampled groups together. The results in Figure 1 show that across the datasets, there exists a spurious group that exhibits higher privacy vulnerability. The largest privacy disparity is observed at ~3% FPR area of Waterbirds, where the samples in the most *spurious group are ~100 times more vulnerable than samples in the non-spurious group.* In CelebA, we continue to observe a significant privacy disparity, with the most spurious group being ~10 times more vulnerable than the least spurious group. In the text dataset MultiNLI, the disparity is milder with ~4 times difference between the most and least vulnerable groups (see Table 1 for the exact numbers at low FPR rate). We have demonstrated the presence of privacy disparity in real-world spurious correlated data. Our results are connected with prior research focused on privacy and fairness (Zhong et al., 2022; Kulynych et al., 2022; Tian et al., 2024), where they also found the presence of privacy disparity between different subpopulations. Surprisingly, we do not observe a spurious privacy leakage in the FMoW dataset, which we investigate in the next section.

> Finding I. *Spurious privacy leakage is present in real-world datasets, where spurious groups can have disproportionately higher vulnerability to privacy attacks than other groups.*

## 3.2 TASK COMPLEXITY AND PRIVACY LEAKAGE

The FMoW dataset, a more complex task with 62 classes compared to Waterbirds or CelebA, exhibits similar privacy vulnerabilities across all the data groups (right-most plot in Figure 1). We show that the number of classes, given a fixed dataset, serves as a proxy for task complexity and analyze how it is related to the *spurious privacy leakage* phenomenon.

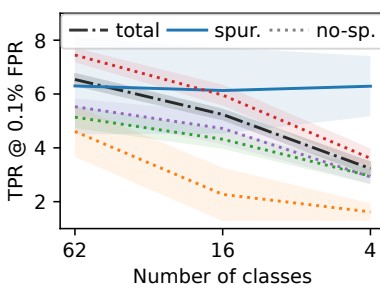
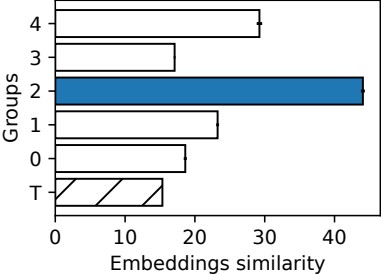

Figure 2: Group privacy disparity increases as the target task simplifies in FMoW. The spurious group (solid line) remains constant while all the other groups become less vulnerable.

Figure 3: Embeddings similarity between FMoW62 and FMoW4 for each group using linear CKA (Kornblith et al., 2019). The most similar group is the spurious group (blue bar).

*Experiment setup.* We partition and group the 62 classes of FMoW into two new datasets with 16 and 4 classes, FMoW16 and FMoW4. We train 16 shadow models for each dataset as in Section 3.1 and use LiRA for privacy analysis. The results are reported over 5 different target models.

The results in Figure 2 show that the average privacy risk over the total dataset decreases as the task progressively simplifies (black dot-dashed line). This observation is consistent with previous works on balanced datasets, such as the increased vulnerability in CIFAR100 compared to CIFAR10 (Shokri et al., 2017; Carlini et al., 2022) and the correlation between higher output dimensions and greater MIA vulnerability in segmentation tasks (Shafran et al., 2021). Interestingly, when zooming in on the dataset, we observe a new phenomenon: *the group privacy disparity emerges between the spurious and non-spurious groups as the task simplifies*. While the leakage for most of the groups drops, the spurious group (Africa) remains consistently vulnerable at 6% across all levels of task complexity. We attribute this phenomenon to the presence of spurious features. We hypothesize that as the task simplifies, models learn *fewer* discriminative features, with spurious groups learning a more *similar* subset of features across tasks compared to non-spurious groups. Firstly, applying PCA on the pre-last layer embeddings to measure the explained variance, we show in Figure 7 that models fit on FMoW4 indeed rely on *fewer* features than FMoW62 and that spurious groups learn *fewer* informative features than non-spurious groups. Then, we use the linear centered kernel alignment (CKA, Kornblith et al. (2019)) to quantify the similarity of embeddings between models fit on FMoW62 and FMoW4. Results in Figure 3 indicate that the spurious group (blue-colored bar) has the highest embedding similarity among all the groups, confirming a higher reuse of similar features.

> Finding II. *Spurious privacy leakage emerges as the task complexity decreases.*

## 4 PRIVACY RISKS OF ROBUST METHODS AND DIFFERENTIAL PRIVACY

We analyze the privacy leakage of models trained using spurious robust training, a family of methods used to suppress spurious correlations. Despite the improvement in group fairness, we observe that the *spurious privacy leakage* phenomenon persists. Lastly, we apply differential privacy and show its limitations in the presence of spurious correlation.

### 4.1 PRIVACY RISKS OF SPURIOUS ROBUST METHODS

Spurious correlations can be suppressed using robust training methods such as group *distributional robust optimization* (DRO) (Sagawa et al., 2019) or *deep feature reweighting* (DFR) (Kirichenko et al., 2022). According to extensive benchmarks in the literature (Izmailov et al., 2022; Yang et al., 2023), DRO and DFR are among the state-of-the-art methods in terms of worst-group accuracy performance. DRO is referred to as an oracle method because it requires a group label to minimize the worst-group error in its objective function (Liu et al., 2021b), and DFR achieves the highest average worst-group accuracy across 12 different spurious datasets across 17 different spurious robust methods (Yang

Table 1: Comparing the attack success rate of different training methods. DFR consistently increases the privacy vulnerability for certain spurious groups across datasets. *TPR results are reported respectively at ~1% and ~3% for the spurious groups 1 and 2 in Waterbirds due to the limited number of samples in the groups. DRO fails to improve the accuracy on FMoW after an extensive grid search, therefore we omit it (see Table 5). The spurious groups are highlighted .

| Data | Group (n) | TPR @ 0.1% FPR (↓) | | | AUROC (↓) | | |
|------|-----------|------|------|------|------|------|------|
| | | ERM | DRO | DFR | ERM | DRO | DFR |
| Waterb. | 0 (1749) | 0.22 ± 0.03 | 0.22 ± 0.03 | 0.22 ± 0.03 | 51.78 ± 0.15 | **51.59 ± 0.16** | 51.64 ± 0.16 |
| | 1 (92)* | **10.87 ± 1.18** | 10.91 ± 1.08 | 11.16 ± 1.20 | 75.07 ± 0.54 | **74.69 ± 0.58** | 75.15 ± 0.52 |
| | 2 (28)* | **30.91 ± 2.81** | 31.06 ± 2.76 | 33.20 ± 2.83 | 85.83 ± 0.76 | **85.54 ± 0.77** | 86.17 ± 0.79 |
| | 3 (528) | **1.73 ± 0.19** | **1.73 ± 0.19** | 1.91 ± 0.20 | 60.52 ± 0.34 | **60.33 ± 0.42** | 60.66 ± 0.33 |
| | T (2397) | 1.16 ± 0.07 | **1.13 ± 0.06** | 1.19 ± 0.06 | 55.44 ± 0.14 | **55.23 ± 0.17** | 55.39 ± 0.15 |
| CelebA | 0 (35814) | 0.53 ± 0.01 | **0.51 ± 0.02** | 0.52 ± 0.02 | 53.12 ± 0.05 | **52.89 ± 0.15** | 53.04 ± 0.11 |
| | 1 (33437) | 0.27 ± 0.01 | **0.26 ± 0.01** | 0.26 ± 0.01 | 50.58 ± 0.05 | **50.48 ± 0.10** | 50.56 ± 0.06 |
| | 2 (11440) | 1.64 ± 0.05 | **1.58 ± 0.06** | 1.62 ± 0.05 | 59.77 ± 0.08 | 59.44 ± 0.26 | **59.36 ± 0.26** |
| | 3 (693) | 4.61 ± 0.50 | **4.56 ± 0.48** | 4.77 ± 0.46 | 80.51 ± 0.21 | **79.95 ± 0.52** | 80.00 ± 0.48 |
| | T (81384) | 0.76 ± 0.01 | **0.73 ± 0.02** | 0.74 ± 0.01 | 53.43 ± 0.04 | **53.22 ± 0.14** | 53.30 ± 0.11 |
| MultiNLI | 0 (14374) | 6.95 ± 0.66 | **6.65 ± 0.61** | 6.78 ± 0.62 | 74.36 ± 0.36 | **74.23 ± 0.40** | 74.26 ± 0.34 |
| | 1 (2789) | **2.03 ± 0.21** | 2.12 ± 0.22 | 2.13 ± 0.21 | **56.81 ± 1.21** | 56.95 ± 1.37 | 56.98 ± 1.36 |
| | 2 (16844) | 5.88 ± 0.42 | **5.73 ± 0.45** | 5.86 ± 0.40 | 72.04 ± 0.31 | **71.93 ± 0.30** | 72.01 ± 0.30 |
| | 3 (380) | 6.14 ± 1.84 | **5.66 ± 1.73** | 6.22 ± 1.84 | 77.41 ± 0.33 | 77.28 ± 0.27 | **77.25 ± 0.30** |
| | 4 (16657) | 5.81 ± 0.25 | **5.67 ± 0.25** | 5.85 ± 0.28 | 75.83 ± 0.15 | **75.64 ± 0.20** | 75.67 ± 0.13 |
| | 6 (498) | 8.26 ± 0.55 | 9.08 ± 1.37 | **7.78 ± 0.57** | 83.70 ± 0.53 | **83.49 ± 0.61** | 83.59 ± 0.57 |
| | T (51542) | 5.95 ± 0.42 | **5.83 ± 0.44** | 5.93 ± 0.42 | 73.44 ± 0.16 | **73.31 ± 0.19** | 73.33 ± 0.13 |
| FMoW | 0 (8904) | **5.14 ± 0.41** | - | 5.22 ± 0.37 | 83.70 ± 0.05 | - | **83.60 ± 0.05** |
| | 1 (17408) | **7.45 ± 0.27** | - | 7.61 ± 0.28 | 85.12 ± 0.07 | - | **84.96 ± 0.08** |
| | 2 (791) | **6.30 ± 1.80** | - | 6.42 ± 1.80 | **81.54 ± 0.22** | - | 81.62 ± 0.24 |
| | 3 (10486) | **5.53 ± 0.31** | - | 5.69 ± 0.32 | 82.85 ± 0.13 | - | **82.74 ± 0.13** |
| | 4 (820) | **4.61 ± 0.95** | - | 5.34 ± 0.88 | 80.37 ± 0.45 | - | **80.26 ± 0.52** |
| | T (38409) | 6.54 ± 0.21 | - | **6.47 ± 0.17** | 84.02 ± 0.05 | - | 83.90 ± 0.03 |

et al., 2023). Therefore, we choose these two methods as representatives for our analysis, where we compare the privacy leakage of the ERM, DRO, and DFR, investigating the possible side effects of suppressing spurious features.

*Experiment setup.* For each dataset and training method, we train the shadow models by following the same LiRA setup as in Section 3. We ensure that models across different training methods use the same subset of data by fixing the random seeds. For privacy evaluation, we train non-overfit models by monitoring the difference between the train-val losses (Yeom et al., 2018).

The performance results in Table 5 show the average and worst-group accuracy of robust training methods for all four datasets. DRO and DFR significantly reduce the difference between train-test WGA by mitigating the influence of spurious features. However, we highlight that relying only on the difference between *average* train-test accuracy can be misleading in detecting overfitting. When comparing two models, the first can have a lower train-test accuracy difference but a higher difference in one of its groups. For example, in CelebA, the ERM method has a lower train-test accuracy difference than DFR (1.3% vs 4.9%) but a higher train-test WGA difference (20.2% vs 5.4%). To truly avoid overfitting, we recommend accounting for the performance disparity across all the groups.

Yeom et al. (2018) demonstrated that overfitting is a sufficient condition for MIA to succeed. In our setting, models trained with spurious robust methods are significantly less overfit across all the data groups (see Table 5). Therefore, we ask: *does mitigating performance disparity also mitigate spurious privacy leakage?* We run the privacy attack with LiRA using ERM trained shadow models and ERM, DRO, and DFR as targets. Our results in Table 1 show the privacy attack success rate for each dataset, group, and training method. Although robust methods successfully mitigate spurious correlations and can mildly reduce the overall privacy at low FPR ("T" rows), they do not affect the privacy disparity. In fact, the leakage for spurious groups is consistently similar for all three training methods across datasets. Our results further extends the findings from Tian et al. (2024), whose analysis is limited to the overall dataset privacy, while we have demonstrated the importance of a per-group privacy audit with *spurious privacy leakage*. Our results may be surprising, but it is known that overfitting is not a necessary condition for MIA to succeed (Yeom et al., 2018).

We provide an alternative explanation of the *spurious privacy leakage* phenomenon by analyzing the memorization score of data, which is related to privacy leakage (Feldman, 2020; Feldman & Zhang, 2020; Carlini et al., 2022). We demonstrate that spurious groups are more vulnerable to MIA due to a higher memorization score compared to other groups (see Appendix B.2 for more details). Our results in Figure 4 show that ERM and DFR share a similar distribution of memorization scores for both spurious and non-spurious groups. DFR only retrains the last layer of the model, but the sample memorization is a phenomenon distributed across various layers (Feldman & Zhang, 2020; Maini et al., 2023), and therefore DFR can hardly affect privacy. For DRO, despite having a similar privacy leakage to ERM (and DFR), it has different memorization scores for spurious groups. Izmailov et al. (2022) demonstrated that DRO acts as DFR by learning, not better features, but a

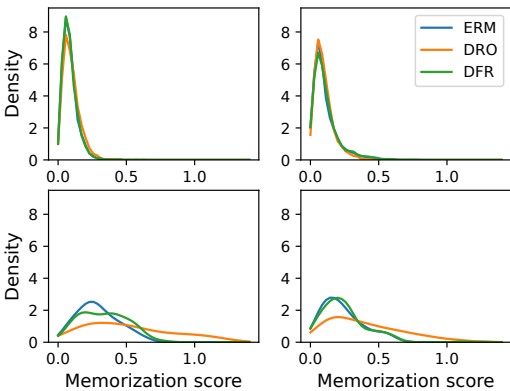

Figure 4: Memorization score per group for each training method. (top) Non-spurious groups have a similar distribution for all the methods. (bottom) Spurious groups have similar ERM and DFR, but DRO has a higher on-average memorization.

better reweighting of a similar set of features, which may explain the similar degree of privacy leakage between DRO, ERM, and DFR.

> **Finding III.** *Spurious robust training methods reduce group performance disparity caused by spurious correlations but fail to address group privacy disparity.*

## 4.2 DIFFERENTIAL PRIVACY FOR SPURIOUS CORRELATIONS

Differential privacy (DP) offers provable privacy guarantees in data protection against membership inference attacks (Dwork, 2006) (Definition 4.1). For neural networks, DP-SGD (Abadi et al., 2016) modifies the SGD optimizer and guarantees the DP properties by adding two steps after gradient computation: clipping the gradient norm with a threshold $C$ and adding random noise to each gradient. In this section, we audit the DP-trained models using LiRA, finding that in practice, spurious groups remain far more vulnerable than other groups even with a low privacy budget.

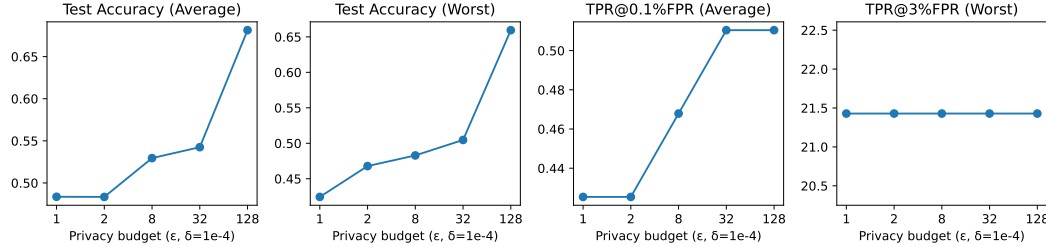

Figure 5: Varying the privacy budget $\epsilon$ for DP-SGD. A tighter budget $\epsilon$ damages the utility for both the average and worst groups, while a higher budget $\epsilon$ achieves a higher utility at the expense of an increased average privacy vulnerability at low FPR. However, we observe no changes in the worst group (right), highlighting the privacy challenges in learning with spurious correlations. See Appendix D.1 for similar results on CelebA and FMoW.

**Definition 4.1** (Differential privacy). A randomized mechanism $\mathcal{M}\colon \mathcal{D} \to \mathcal{R}$ satisfies $(\epsilon, \delta)$-differential privacy if for any two datasets differing by a single data point $D, D' \in \mathcal{D}$ and for any subset of outputs $S \subseteq \mathcal{R}$ it holds that $\Pr[\mathcal{M}(D) \in S] \leq e^{\epsilon} \Pr[\mathcal{M}(D') \in S] + \delta$.

*Experiment setup.* We use the fastDP library (Bu et al., 2023) to train CNext-T (Liu et al., 2022b) target models with full batch DP training (De et al., 2022; Panda et al., 2024). The model selection

uses a grid search with lr in [1, 1e-1, 1e-2, 1e-3], $\epsilon$ in [1, 2, 8, 32, 128], and $\delta$ = 1e-4. We observed that weight decay and the cosine scheduler make the optimization unstable and remove them as in Panda et al. (2024). Each model is trained up to 100 epochs and we choose the checkpoint with the best validation WGA. The privacy attack is run using LiRA with 32 CNext-T ERM shadow models. For the evaluation, the privacy budget is reported as $(\epsilon, \delta)$ where $\epsilon$ is the desired privacy guarantee and $\delta$ is the failure probability (see Appendix D for more details).

We investigate the possibility of mitigating the *spurious privacy leakage* by controlling the utility-privacy tradeoff using differential privacy under MIA. The results in Figure 5 show the performance of the full-batch trained target models across four metrics, covering the average and worst-group utility and privacy metrics across a range of values for privacy budget $\epsilon$. Bagdasaryan et al. (2019) show that DP training increases the group performance disparity, hurting the performance of small groups. However, similarly to Panda et al. (2024), we observe that DP training *with large batch size* can mitigate the group performance disparity, improving the worst-group performance. In particular, when using a tight budget of $\epsilon = 1$, the model's performance is drastically affected, reducing to mere random chance (~50%). When relaxing the privacy budget with $\epsilon = 128$, the average and worst-group utility improves from ~50% to ~65%, but also the average privacy vulnerability across the whole dataset increases, from ~0.43% to ~0.51% (third plot of Figure 5). The most concerning result lies within the worst-performing spurious group, whose privacy vulnerability remains constant at ~21.5% across various budgets $\epsilon$.

> Finding IV. *Differential privacy fails to mitigate the privacy vulnerability of spurious groups.*

## 5    ARCHITECTURE INFLUENCE ON SPURIOUS CORRELATIONS AND PRIVACY

Prior privacy works mostly focused on settings with ResNet-like architecture. However, modern architectures have different components that impact feature learning (e.g. masking from He et al. (2022) or attention from Dosovitskiy et al. (2021)). We include models that are sufficiently "diverse". For example, we compare models from different families (convolution and transformers), with different pretraining strategies (supervised and self-supervised), or released at different times (e.g. ResNet and its successor ConvNext).

*Experiment setup.* All the models used are pretrained using the state-of-the-art recipe on the ImageNet1K dataset from the timm library. We compare ResNet50 (He et al., 2016), BiT-S (Kolesnikov et al., 2020), CNext-T (Liu et al., 2022b), CNextV2-T (Woo et al., 2023), ViT (Dosovitskiy et al., 2021), Swin-T (Liu et al., 2021c), DeiT-S (Touvron et al., 2022), and Hiera (Ryali et al., 2023). To ensure a fair comparison, all the models have a similar number of parameters, between 20M to 30M. We train 16 shadow models for each architecture as in Section 3 while monitoring the train-validation loss to prevent overfitting. Table 7 reports the details about our grid search and summarizes the best hyperparameters for each model. The results are averaged over 16 seeds.

**Are ViTs more spurious robust than CNNs?** Ghosal & Li (2024) claim that vision transformers are more robust than convolutional models. We revisit the statement, showing that under the same grid search, the best transformer performs similarly to the convolutional model (see Table 2, Swin-T is similar to CNextV2-T). We highlight that using the same architecture, DeiT-S/16, our results achieve higher test WGA (+4.7% absolute) compared to the one reported by Ghosal & Li (2024) (Table 4) while using only half of the Waterbirds training data. Moreover, Ghosal & Li (2024) unfairly compare BiT with ViT/DeiT, where the latter is pretrained with a more complex optimization recipe (RandAug+Mixup+CutMix+LabelSmoothing+StochasticDepth+LayerScale) but BiT is not. Our benchmark fairly allocates a fixed compute budget for all the models.

> Finding V. *Vision transformers are not more spurious robust than convolutional models under a fair experimental setup (revisiting Ghosal & Li (2024)).*

**Pretraining recipe and architecture matter for privacy auditing.** Prior works audit the privacy of different architectures on well-balanced datasets such as CIFAR or ImageNet (Carlini et al., 2022; Liu et al., 2022a). We use a new auditing setup to compare state-of-the-art architectures, showing the importance of architecture choice for privacy auditing on non-balanced datasets, Waterbirds. Using

Table 2: Target model architecture accuracy on Waterbirds dataset. Modern architectures are better at mitigating spurious correlation than older ones, with no significant worst-group accuracy difference between the best convolutional and transformer-based architectures.

| Model | Train Acc. | Test Acc. | Test WGA |
|---|---|---|---|
| ResNet50 | $96.94 \pm 0.03$ | $81.08 \pm 0.25$ | $34.42 \pm 0.43$ |
| BiT-S | $96.73 \pm 0.07$ | $79.90 \pm 0.21$ | $42.37 \pm 0.89$ |
| CNext-T | $97.47 \pm 0.04$ | $83.36 \pm 0.32$ | $47.73 \pm 0.97$ |
| CNextV2-T | $98.33 \pm 0.09$ | $\mathbf{83.96 \pm 0.22}$ | $51.90 \pm 1.38$ |
| ViT-S | $97.46 \pm 0.07$ | $80.76 \pm 0.20$ | $43.68 \pm 0.60$ |
| Deit3-S | $97.27 \pm 0.05$ | $83.66 \pm 0.15$ | $51.46 \pm 0.49$ |
| Swin-T | $\mathbf{98.43 \pm 0.07}$ | $83.72 \pm 0.36$ | $\mathbf{52.06 \pm 1.30}$ |
| Hiera-T | $98.27 \pm 0.09$ | $82.60 \pm 0.37$ | $43.58 \pm 0.83$ |

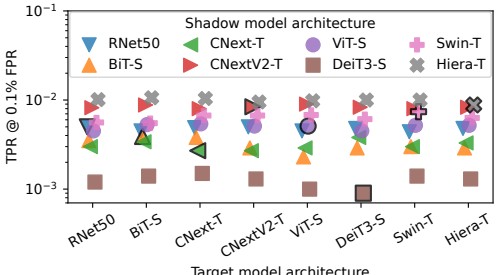

Figure 6: Varying the target and shadow model architecture on the entire Waterbirds dataset. The most successful attack is not when the shadow correctly guesses the target architecture.

LiRA, we run the attack through all the permutations of shadow/target architecture pairs, resulting in 64 different configurations. Our results in Figure 6 show that when we fix the shadow architecture and vary the target, there is no particular architecture that is more resistant to others. On the other hand, when fixing the target architecture while varying the shadow, we observe that self-supervised pretrained models, Hiera and CNextV2, consistently achieve the strongest attack across all the targets. Both self-supervised models are pretrained on ImageNet1k with masked autoencoders (He et al., 2022), which may enhance the attack success and presents an interesting direction for future work. Moreover, despite sharing the same architecture with ViT but differing in optimization recipe, DeiT is consistently ranked as the weakest attack. These observations suggest that *the pretraining optimization can greatly influence the attack success of shadow models.* Lastly, prior works reported that the most successful attack happens when the shadow matches the target architecture (Carlini et al., 2022; Liu et al., 2022a). In our setup with spurious data, we do not observe the same pattern. In Figure 6, the black-outlined markers represent the match between shadow and target architectures, which is consistently a suboptimal attack.

> Finding VI. *The best choice for the shadow architecture does not always match the target.*

## 6  RELATED WORK

The intersection of privacy and ML safety topics has been the focus of several studies. Wang et al. (2020) demonstrated how pruning can mitigate privacy attacks, Shokri et al. (2021) explored the connection between privacy and explainability, and Song et al. (2019) found that adversarial training can increase privacy leakage. However, Li et al. (2024) reported contradictory adversarial training findings when using a better evaluation guideline (Carlini et al., 2022), highlighting the importance of revisiting prior claims. In our work, we investigate the privacy risk of real-world spurious correlated datasets, which is also related to fairness machine learning, where the goal is to ensure equal performance across groups.

Prior work has extensively explored the intersection of privacy and fairness, demonstrating that subpopulations often exhibit varying levels of privacy risk (Tian et al., 2024; Zhong et al., 2022; Kulynych et al., 2022). For example, Tian et al. (2024) showed that fairness methods can mildly mitigate MIA risks when considering aggregate metrics in binary classification tasks, consistent with our findings in Section 3 (see "T" in Table 4). Instead, our work focuses on spurious correlation and extends the fairness results by conducting a per-group analysis, revealing that spurious groups remain highly vulnerable despite improvements in overall metrics. Kulynych et al. (2022) and Zhong et al. (2022) also explored the concept of privacy disparity, focusing on synthetic or tabular datasets and hypothesizing that group fairness improvements or differential privacy (DP) could mitigate these disparities. In Section 4.1, we show that both approaches fail to address privacy disparities in real-world datasets with spurious correlations. Nevertheless, in Section 4.2, we observe that large batch DP-SGD training can even improve fairness, consistent with by Panda et al. (2024), but contradicting with Bagdasaryan et al. (2019) and Farrand et al. (2020) using smaller batch sizes.

Lastly, Yang et al. (2022) examined the privacy risks associated with a spurious correlated toy dataset (MNIST with color perturbations) using a suboptimal evaluation. In contrast, our results use state-of-the-art methods to evaluate real-world datasets directly from the spurious correlation literature, solidifying the past findings and providing additional insights into the effectiveness of mitigation methods (robust methods and differential privacy in Section 4.1) and the impact of model architecture choices (Section 5).

## 7 CONCLUSION

Our findings expose critical privacy concerns when training neural networks on datasets with spurious correlations. We demonstrate the existence of *spurious privacy leakage* in real-world datasets, where spurious data groups are more vulnerable to privacy attacks than non-spurious groups. This phenomenon is masked by aggregate metrics, emphasizing the need for privacy audits that include fine-grained and group-level analyses to ensure both performance and privacy fairness. Additionally, we point out the limitations of the current methods: *neither spurious robust training nor using differential privacy mitigate spurious privacy leakage in practice*. Lastly, we revisit prior work on the relationship between architecture, spurious correlations, and privacy, providing insights that revisit and complement existing research. Our contributions identify overlooked challenges and present opportunities for future research on privacy and spurious correlations.

**Impact.** Our findings have significant implications for the machine learning communities concerned with bias, fairness, and security. Understanding the connection between spurious correlations and privacy is important for assessing the risks within data-sensitive domains. In particular, we suggest practitioners working in these areas to verify the presence of privacy disparities.

**Limitations.** Our results are based on an attack-based evaluation rather than analyzing the worst-case guarantees. While our approach is more practical and provides empirical evidence, it is limited to the choice of experiment settings, which include two robust training methods, eight model architectures, and four real-world datasets.

**Reproducibility.** We have made our code publicly available through an anonymized repository (Section 1). Details for each experiment setup are presented in the respective sections, along with the corresponding grid search and the best hyperparameters in the appendix (see Tables 3 and 7). The spurious datasets are presented in the Appendix A.

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

APPENDIX

We report the dataset details, additional results on group privacy disparity, a comparison of different membership inference attack methods, define and show the memorization score for each dataset, and more results on differential privacy and model architectures.

## A DATASET

**Waterbirds** Sagawa et al. (2019). Vision dataset where the task is to classify whether landbird or waterbird. The background is the spurious feature represented as water or land background. The presence of the spurious features induces four data groups: landbird on land background, landbird on water background, waterbird on water background, and waterbird on land background. The groups have respectively 3498, 184, 1057, and 56 samples. Therefore, the type of bird is spurious correlated with the same type of background.

**CelebA** Liu et al. (2014). Vision dataset where the task is to classify whether a celebrity is a male or female. The hair color is the spurious features represented as dark or blonde hair. The presence of spurious features induces four data groups: female with blonde hair, female with dark hair, male with dark hair, and male with blonde hair. The groups have respectively 71629, 66874, 22880, and 1387 samples. Therefore, blonde hair is spurious correlated with female celebrities.

**FMoW** Koh et al. (2021). Vision dataset where the task is to identify between 62 classes the type of land usage, e.g. hospital, airport, single or multi-use residential area. The geographical location is the spurious feature representing the continents: Asia, Europe, Africa, Americas, and Oceania. The groups have respectively 17809, 34816, 1582, 20973, and 1641 samples whereas the African countries have the majority of samples as single-use residential areas (36%). Therefore, samples collected from Africa are spurious correlated with the single-unit residential areas. Moreover, the test set presents a distribution shift with samples collected from different years.

**MultiNLI** Williams et al. (2017). Text dataset where the task is to identify the relationship between two pairs of text as a contradiction, entailment, or neither. The negation is the spurious feature usually found in the contradiction class. The presence of the spurious feature induces six data groups: contradiction without negation, contradiction with negation, entailment without negation, entailment with negation, neutral without negation, and neutral with negation. The groups have respectively 57498, 11158, 67376, 1521, 66630, and 1991 samples. Therefore, samples with the spurious feature negation are correlated with the contradiction class.

## B SPURIOUS PRIVACY LEAKAGE

We report additional technical details related to Section 3 and include additional results: comparing different membership inference attacks on spurious data, demonstrating how memorization of spurious data causes higher privacy leakage.

*Hyperparameters.* For Section 3, we apply grid search to find the best hyperparameters for each dataset. For Waterbirds and CelebA we search the learning rate between [1e-3, 1e-4] and weight decay [1e-1, 1e-2, 1e-3]. For FMoW the learning rate [1e-3, 3e-3, 1e-4, 3e-4], weight decay [1e-1, 1e-2, 1e-3], and epochs [20, 30, 40]. For MultiNLI the learning rate [1e-5, 3e-5], weight decay [1e-5, 1e-4]. The best hyperparameters are reported at Table 3.

Table 3: Hyperparameters used to train shadow models for each dataset. Adapted from the hyperparameters of Izmailov et al. (2022). Since we trained the models using LiRA algorithm with 50% of the total dataset, we had to grid search and validate on the validation set.

| Data | Optim | Batch size | LR | WD | Epochs | C |
|------|-------|-----------|------|------|--------|---|
| Waterbirds | SGD | 32 | 1e-3 | 1e-2 | 100 | 1 |
| CelebA | SGD | 32 | 1e-3 | 1e-2 | 20 | 5 |
| FMoW | SGD | 32 | 3e-3 | 1e-2 | 20 | 1 |
| MultiNLI | AdamW | 16 | 1e-5 | 1e-4 | 5 | 8 |

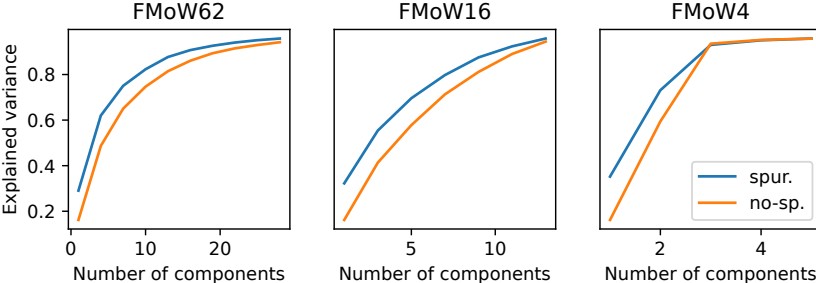

Figure 7: Explainable variance for models trained on different data complexity. We apply the PCA to the embeddings of models trained on FMoW, FMoW16, and FMoW4. The dataset with 4 classes needs only ~3 components to explain the 0.90% of the variance, much lower compared to FMoW16 and FMoW62 classes that require respectively ~15 and ~20 components. Moreover, spurious groups consistently need less number of components than non-spurious groups across data complexity, indicating fewer features are learned.

### B.1 MEMBERSHIP INFERENCE ATTACKS COMPARISON

Most of the previous MIAs are limited by the assumption that all the samples have the same level of importance (or hardness) (Yeom et al., 2018; Shokri et al., 2017), which is incorrect since natural data follow a long-tail distribution (Feldman, 2020). We compare three different state-of-the-art MIAs and show that the phenomenon of *spurious privacy leakage* exists regardless of the attack used. We use two different versions of LiRA (Carlini et al., 2022), online and offline, and TrajMIA (Liu et al., 2022a). The results in Table 4 show that all the methods successfully reveal the disparity on Waterbirds, and LiRA online is the strongest attack on vulnerable groups.

Table 4: Comparing the attack success rate of different membership inference attacks on ERM models trained with Waterbirds. All the methods can be used to identify the privacy disparity, but LiRA poses a greater risk for more vulnerable spurious groups. *TPRs are reported at ~1% and ~3% for groups 1 and 2 respectively due to their limited sample size. The spurious groups are highlighted .

| | TPR @ 0.1% FPR (↑) | | | AUROC (↑) | | |
| Group | LiRA | LiRA (offline) | TrajMIA | LiRA | LiRA (offline) | TrajMIA |
|---|---|---|---|---|---|---|
| 1 | $0.22 \pm 0.03$ | $0.14 \pm 0.02$ | $\mathbf{1.67 \pm 3.27}$ | $51.78 \pm 0.15$ | $49.97 \pm 0.22$ | $\mathbf{58.20 \pm 3.42}$ |
| 2* | $\mathbf{10.87 \pm 1.18}$ | $5.39 \pm 0.78$ | $3.18 \pm 0.47$ | $\mathbf{75.07 \pm 0.54}$ | $61.32 \pm 1.01$ | $70.28 \pm 1.22$ |
| 3* | $\mathbf{30.91 \pm 2.81}$ | $18.98 \pm 2.13$ | $14.60 \pm 1.69$ | $\mathbf{85.83 \pm 0.76}$ | $69.50 \pm 1.67$ | $\mathbf{86.16 \pm 2.55}$ |
| 4 | $1.73 \pm 0.19$ | $0.83 \pm 0.11$ | $\mathbf{6.57 \pm 0.59}$ | $60.52 \pm 0.34$ | $53.63 \pm 0.42$ | $\mathbf{72.40 \pm 2.31}$ |
| T | $1.16 \pm 0.07$ | $0.44 \pm 0.04$ | $\mathbf{1.68 \pm 0.00}$ | $55.44 \pm 0.14$ | $51.43 \pm 0.16$ | $\mathbf{74.74 \pm 0.00}$ |

### B.2 MEMORIZATION SCORE OF SPURIOUS GROUPS

Feldman (2020) introduced the notion of label memorization (Definition B.1) as the difference in the label of a model trained with or without $\boldsymbol{x}$. We use the models from the LiRA algorithm from Section 3.1 to approximate the memorization score. Carlini et al. (2022) proposed the privacy score $d = |\mu_{\text{in}} - \mu_{\text{out}}| / (\sigma_{\text{in}} + \sigma_{\text{out}})$ to measure the difference between the loss distributions coming from IN and OUT shadow models of LiRA. Note that both mem(.) and $d$ measure the difference between two probability distributions conditioned on $D$ and $D \setminus \{i\}$ but with a different level of granularity; label memorization is coarser than $d$ and collapses the whole distributions to a single scalar, the probability of outputting the correct label.

**Definition B.1** (Label memorization). Label memorization is the difference in the output label of a model $f \sim \mathcal{A}(\mathcal{D})$ fit on the dataset $D$ with or without a specific data point $(\boldsymbol{x}_i, \boldsymbol{y}_i) \sim D$. Formally, $\text{mem}(\mathcal{A}, D, i) = \left| \text{Pr}_{f \sim \mathcal{A}(D)} (f(\boldsymbol{x}_i) = \boldsymbol{y}_i) - \text{Pr}_{f \sim \mathcal{A}(D \setminus \{i\})} (f(\boldsymbol{x}_i) = \boldsymbol{y}_i) \right|$

We compute $d$ for each data point and use a Gaussian kernel density estimator to fit each group. The results in Fig. 8 show the estimated frequency of the memorization score for the whole dataset divided per group. We observe that the spurious groups have, on average, higher memorization scores

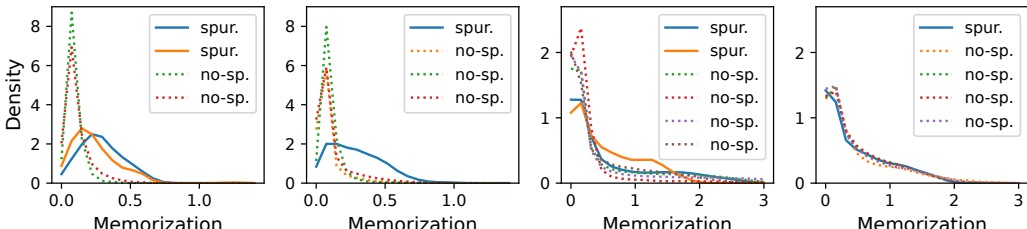

Figure 8: Memorization score divided per group on Waterbirds, CelebA, MultiNLI, and FMoW respectively. Spurious correlated groups (solid lines) have on average a higher memorization score than non-spurious groups, which indicates that models treat spurious groups as atypical examples. In the FMoW dataset, all the groups have similar levels of memorization.

Table 5: Evaluating the different training methods across datasets. DRO and DFR consistently mitigate spurious features by reducing the gap between train-test WGA compared to ERM. After an extensive grid search, DRO fails to improve the validation WGA on FMoW, therefore we omit it.

|  | Model | Train Acc. (↑) | Test Acc. (↑) | Diff. Acc. (↓) | Train WGA (↑) | Test WGA (↑) | Diff. WGA (↓) |
|---|---|---|---|---|---|---|---|
| Waterb. | ERM | **97.16 ± 0.11** | 81.12 ± 0.35 | 16.0 | 50.18 ± 2.70 | 34.30 ± 1.27 | 15.8 |
|  | DRO | 96.16 ± 0.23 | **86.42 ± 0.38** | 9.7 | **93.73 ± 0.44** | 78.12 ± 0.84 | 15.6 |
|  | DFR | 92.63 ± 1.13 | 85.98 ± 0.60 | **6.7** | 85.81 ± 1.95 | **77.67 ± 2.13** | **8.2** |
| CelebA | ERM | **97.12 ± 0.03** | **95.82 ± 0.06** | 1.3 | 62.81 ± 1.82 | 42.67 ± 0.62 | 20.2 |
|  | DRO | 94.47 ± 0.05 | 93.23 ± 0.21 | **1.2** | **91.84 ± 0.36** | **86.11 ± 0.89** | 5.7 |
|  | DFR | 95.43 ± 0.14 | 90.52 ± 0.22 | 4.9 | 89.46 ± 0.36 | 84.00 ± 0.60 | **5.4** |
| MultiNLI | ERM | **97.26 ± 0.04** | 80.74 ± 0.04 | 16.5 | **91.43 ± 0.79** | 61.76 ± 0.28 | 29.7 |
|  | DRO | 89.69 ± 0.09 | 78.76 ± 0.07 | **10.9** | 85.34 ± 0.23 | **72.96 ± 0.66** | **12.4** |
|  | DFR | 96.36 ± 0.14 | **79.17 ± 0.06** | 17.2 | 90.84 ± 0.11 | 71.33 ± 0.13 | 19.5 |
| FMoW | ERM | 91.58 ± 0.04 | **50.85 ± 0.08** | **40.7** | **90.84 ± 0.06** | 31.04 ± 0.20 | 59.8 |
|  | DRO | - | - | - | - | - | - |
|  | DFR | **91.20 ± 0.38** | 48.62 ± 0.09 | 42.6 | 88.57 ± 0.55 | **32.44 ± 0.34** | **56.1** |

compared to non-spurious groups (except for FMoW as in Fig. 1). The increase can be attributed to the presence of spurious features, which turn typical examples into atypical ones that the model has to memorize. A higher memorization score is known to be linked to a higher vulnerability under privacy attacks (Feldman, 2020), which matches what we observed previously.

## C ROBUST TRAINING

We report additional technical details related to Section 4.1 and include an additional result analyzing the privacy side effect of choosing L2 vs L1 regularization in DFR.

*Hyperparameters.* We use the same hyperparameters as in Table 3. Robust training DRO requires an extra hyperparameter C. For Waterbirds and CelebA we tune C within [0, 1, 2, 3, 4], for FMoW [0, 1, 2, 4, 8, 16], and for MultiNLI [0, 1, 2, 4, 8, 16]. For DFR, we do not use the validation set for retraining but use a group-balanced subset sampled from the training set. This allows a fairer comparison with other methods by not exploiting additional data, and it is also necessary for a fair privacy analysis since adding extra data invalidates the membership inference comparison.

*Experiment setup.* For the LiRA attack, we train 32 ERM shadow models for Waterbirds and CelebA and 16 ERM shadow models for FMoW and MultiNLI. We also train 5 DRO and DFR models for Waterbirds and CelebA, and 32 DRO and DFR models for FMoW and MultiNLI. We use the online version of LiRA with a fixed variance for all the attacks to audit the privacy level. Table 1 reports the mean and standard error of using the ERM trained shadow models to attack 32 target models for each training type of Waterbirds and CelebA, and 5 for FMoW and MultiNLI.

We found DRO to be unstable on more complex datasets such as FMoW, where it fails to improve the validation WGA even after an extensive hyperparameter grid search. While for DFR, despite its simplicity and effectiveness compared to DRO, we find that it can slightly increase the vulnerability

of spurious groups. However, by simply changing DFR's regularization from L1 to L2 norm, we achieve an accuracy-privacy tradeoff reducing the vulnerability to the same level as ERM at the cost of a lower WGA (77.67% to 73.00%) (see Appendix C.1).

## C.1 DFR WITH L2 REGULARIZATION

DFR with the L1 regularization achieves the best performance measured with WGA. The L1 regularization encourages sparsity of the last-linear layer, concentrating most of the weights to 0. Kirichenko et al. (2022) showed that using L2 regularization leads to suboptimal results in terms of WGA performance. Additionally, we find that L2 leads to an accuracy-privacy tradeoff where it slightly increases privacy protection against MIA. We compare DFR trained with L2 and the default L1 regularization, finding that L2 regularization achieves a lower 83.65% test accuracy compared to 85.96% of L1, and also a lower WGA 73.00% compared to 77.67%. However, in Table 6, we observe that by using L2 regularization, the privacy vulnerability is reduced to a similar level of ERM.

Table 6: Comparing DFR L2 and L1 regularization under the LiRA attack with 32 ERM shadow models trained on Waterbirds. The results are averaged over 32 target models.

| | TPR @ low% FPR | | |
|---|---|---|---|
| Group (n) | ERM | DFR L1 | DFR L2 |
| 0 (1749) | $0.22 \pm 0.03$ | $0.22 \pm 0.03$ | $0.22 \pm 0.03$ |
| 1 (92) | $10.87 \pm 1.18$ | $11.16 \pm 1.20$ | $\mathbf{10.84 \pm 1.16}$ |
| 2 (28) | $30.91 \pm 2.81$ | $33.20 \pm 2.83$ | $\mathbf{30.52 \pm 2.70}$ |
| 3 (528) | $1.73 \pm 0.19$ | $1.91 \pm 0.20$ | $\mathbf{1.67 \pm 0.21}$ |
| T (2397) | $1.16 \pm 0.07$ | $1.19 \pm 0.06$ | $\mathbf{1.10 \pm 0.07}$ |

## D DIFFERENTIAL PRIVACY

**Definition D.1** (Differential privacy). A randomized mechanism $\mathcal{M} \colon \mathcal{D} \to \mathcal{R}$ satisfies $(\epsilon, \delta)$-differential privacy if for any two datasets differing by a single data point $D, D' \in \mathcal{D}$ and for any subset of outputs $S \subseteq \mathcal{R}$ it holds that

$$\Pr[\mathcal{M}(D) \in S] \leq e^\epsilon \Pr[\mathcal{M}(D') \in S] + \delta.$$

where $\epsilon \geq 0$ and $\delta \geq 0$ are privacy parameters. A higher privacy budget $(\epsilon, \delta)$ results in a better utility but lower protection, while a lower privacy budget guarantees the opposite. The privacy budget $(\epsilon, \delta)$ in DP-SGD Abadi et al. (2016) is controlled by the hyperparameters noise level $\sigma$ added to the gradient and clipping threshold $C$ to clip the maximum norm of the gradient. A higher level of noise leads to a lower privacy budget, and the clipping influences the amount of possible noise to add.

We use the same hyperparameters reported in Table 7 to train the ConvNext. We do not use the ResNet as the batch normalization is not compatible with DP-SGD. Specifically, batch normalization creates a dependency between samples in a batch which is a privacy violation. To audit the privacy level, 32 ERM shadow models are trained using the ConvNext architecture, and target models are trained using the same setting in addition to the DP's privacy budget.

## D.1 ADDITIONAL RESULTS

We complement Section 4.2 with additional results.

*Experiment setup.* We train DP-SGD target models on CelebA and FMoW for 50 epochs using batch size 1024 and grid searching the lr in [1e-1, 1e-2], $\epsilon$ in [1, 2, 8, 32, 128], and $\delta$ of 1e-5. We use the Opacus library due to errors with fastDP. We reuse the shadow models trained in Section 3 for LiRA.

Consistent with our findings for Waterbirds Section 4.2, low $\epsilon$ (=1, 2) hinders the generalization, while high $\epsilon$ better mitigates spurious correlations. However, in terms of privacy vulnerability, all models across different privacy budget levels exhibit similar levels of vulnerability, highlighting the challenges of applying DP while retaining the utility of spurious data.

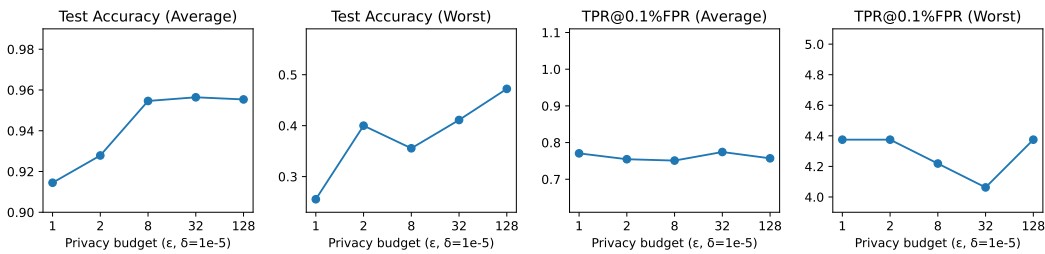

Figure 9: Varying the $\epsilon$ for target models trained on CelebA with DP-SGD

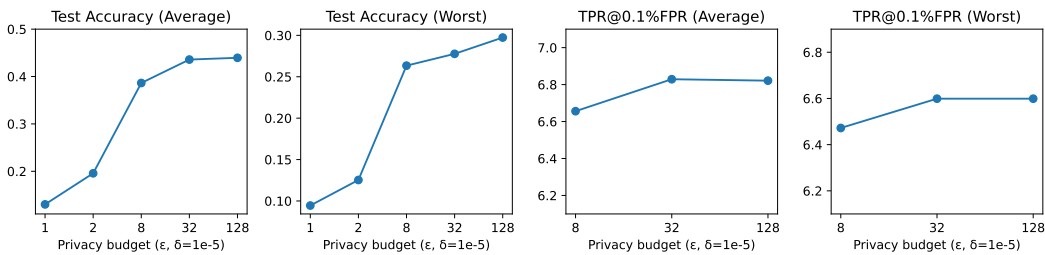

Figure 10: Varying the $\epsilon$ for target models trained on FMoW with DP-SGD

### D.2 MINI-BATCH DIFFERENTIAL PRIVACY

We combine the robust training method DRO with DP, aiming to train a group fair model with a provable privacy guarantee.

*Experimet setup.* We use the Opacus library (Yousefpour et al., 2021) to train private target models with a batch size of 32 using different privacy budgets $\epsilon$ on Waterbirds. All the target models are trained with the same subset of data to ensure a fair comparison. The target architecture is ConvNext-T pretrained on ImageNet1k with layer normalization (instead of batch normalization) and therefore is compatible with DP-SGD. We run the attack using LiRA with 32 ConvNext-T shadow models trained with the same method as in Section 3.

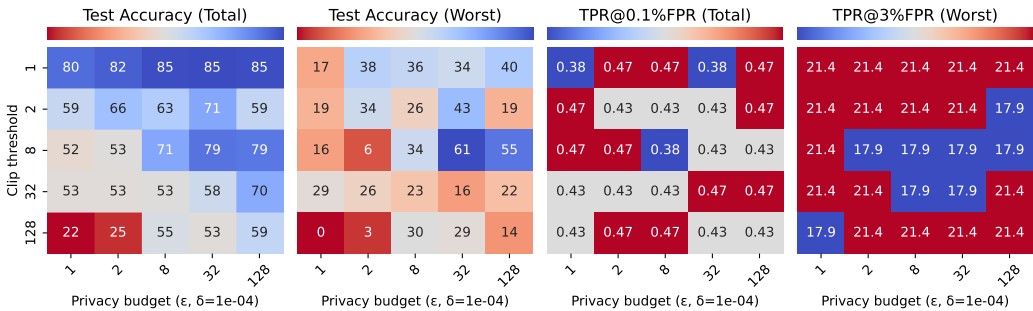

Figure 11: Combining DP-SGD with spurious robust method DRO on Waterbirds. Reducing the privacy budget $\epsilon$ reduces the utility in the worst group (second column). Even with a tight budget the membership inference attack still succeeds for the spurious group (fourth column), showing the practical limitations of differential privacy.

We observe that using a high privacy budget $\epsilon$ it is possible to maintain a higher WGA. However, as the privacy budget is tightened, the average and worst-group utility drops. Moreover, we observe that all the models exhibit similar levels of vulnerability across all the budgets, in particular for the worst group (right-most figure in Fig. 11).

Bagdasaryan et al. (2019) empirically showed that training with mini-batched DP can further increase the unfairness of already unfair data, i.e. "the poor become poorer". We observe the same and

additionally in Fig. 11. Despite significantly reducing the unfairness by training with spurious robust methods (Table 5), tight-budget DP still heavily limits the performance of the spurious group. Our results suggest that simply reducing the unfairness with better algorithms may not be sufficient. We suggest practitioners focus on minimizing the bias present in the dataset.

# E  ARCHITECTURE INFLUENCES

We report additional technical details related to Section 5.

*Experimet setup.* For each model, we use the same computational budget by performing the grid search with the learning rate [1e-1, 1e-2, 1e-3, 1e-4] and the weight decay in [1e-1, 1e-2, 1e-3] and choose the best performing based on the validation set. For privacy analysis, we avoid overfitting by tuning the number of training epochs to stop the training before reaching 100% training accuracy (Carlini et al., 2022). The best hyperparameters are reported at Table 7.

Table 7: Final hyperparameters used for training the various architecture in Section 5.

| Model | Params (M) | Batch size | LR | WD | Epochs |
|-------|-----------|-----------|------|------|--------|
| ResNet50 | 23.5 | 32 | 1e-3 | 1e-2 | 100 |
| BiT-S | 23.6 | 32 | 1e-4 | 1e-2 | 10 |
| CNext-T | 27.8 | 32 | 1e-3 | 1e-2 | 10 |
| CNextV2-T | 27.8 | 32 | 1e-3 | 1e-2 | 5 |
| ViT-S | 21.6 | 32 | 1e-4 | 1e-2 | 10 |
| Deit3-S | 21.6 | 32 | 1e-4 | 1e-2 | 10 |
| Swin-T | 27.5 | 32 | 1e-3 | 1e-2 | 10 |
| Hiera-T | 27.1 | 32 | 1e-4 | 1e-2 | 20 |

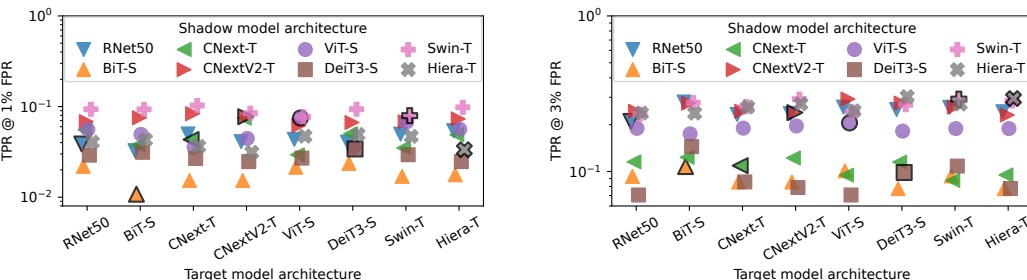

Figure 12: Varying the target and shadow model architecture on the whole Waterbirds dataset for the spurious groups. The least spurious robust architecture (ResNet50) consistently achieves a higher attack success rate on all the target architectures, and the opposite phenomenon happens for the most spurious robust architecture (DeiT3-S).

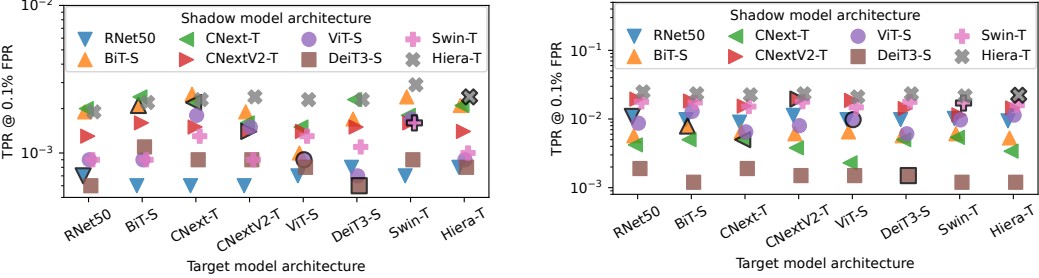

Figure 13: Varying the target and shadow model architecture on the whole Waterbirds dataset for the non-spurious groups. There is no trend for the most typical group (left), while the other group follows the trend as in spurious groups.

## F  COMPUTE RESOURCES

All the experiments are run on our internal cluster with the GPU Tesla V100 16GB/32GB of memory. We give an estimate of the amount of compute required for each experiment. For Section 3, we trained 96 shadow models for Waterbirds and CelebA, and 48 for FMoW and MultiNLI which took ~300 hours of computing. Moreover, we trained 16 shadow models for FMoW4 and FMoW16 which took another ~50 hours. For Section 5, we trained in total 128 shadow models on Waterbirds averaging around ~100 hours. Lastly for Section 4.2, we trained 32 ConvNext-t shadow models and 5 target models for about ~50 hours. The full research required additional computing for hyperparameter grid searches, in particular for differential privacy training which is known to be difficult to optimize.

