# OpenReview forum: "Spurious Privacy Leakage in Neural Networks"
_ICLR.cc/2025/Conference — Submitted to ICLR 2025_

### Official Review · Reviewer_9gk7 · 2024-10-22

**Soundness:** 2
**Presentation:** 2
**Contribution:** 2
**Rating:** 3
**Confidence:** 4

**Summary:**

The paper examines how bias in neural networks affects privacy vulnerabilities, particularly focusing on spurious correlations. The paper has several findings after experiments.

**Strengths:**

The spurious privacy leakage in neural networks is an interesting topic and the paper has a good format emphasizing each finding after some experiments.

**Weaknesses:**

1. Current research has much more literature than what the paper presented in the related work section.
2. The authors should explain why these datasets are chosen and not the other datasets.
3. In Sec 3.1, the authors mention that "The largest privacy disparity is observed at ~3% FPR area of Waterbirds, where the samples in the most spurious group are ~100 times more vulnerable than samples in the non-spurious group." Where do "3%" and "100 times" come from? The figure does not clearly show this and the paragraph does not explain this.
4. The claim of "spurious groups can be up to 100 times more vulnerable to privacy attacks than non-spurious groups" is just a result from one data point. This is exaggeration and should not include in the abstract.
5. While the paper presents experiments for its findings, the limited number of experiments conducted for each conclusion raises questions about results. Additional experiments are needed for each finding.
6. Finding V claims that they draw the conclusion under a fair experimental setup. However, the authors use different hyperparameter settings for different models, which is not a fair comparison.

**Questions:**

See weaknesses.

---

> ### Author Response · Authors · 2024-11-17
>
> We thank the reviewer for the feedback. Below we address each point:
>
> > 6) Finding V claims that they draw the conclusion under a fair experimental setup. However, the authors use different hyperparameter settings for different models, which is not a fair comparison.
>
> The reviewer expects us to use the same hyperparameter settings to compare different models. However, the process of benchmarking in Machine Learning follows a different procedure. For each model, a grid search of hyperparameters is used and the best set of hyperparameters is chosen based on a held-out validation dataset (see section 11.4.3 of [1] or section 8.5 of [2]). We have followed exactly this approach described in Appendix E.
>
> > 5) [...] Additional experiments are needed for each finding
>
> See overall comment 2).
>
> > 1) Current research has much more literature than what the paper presented in the related work section
>
> See overall comment 1).
>
> > 2) The authors should explain why these datasets are chosen and not the other datasets
>
> We clarify the section 3.1 by adding:
>
> Experiment setup. We choose the datasets that are commonly used by the spurious correlation community (see Table 2 of [3]): Waterbirds, CelebA, FMoW, and MultiNLI. These datasets contain real-world spurious correlations, diverse modalities, and different target complexity. Moreover, to the best of our knowledge, we are the first to study subgroup MIA attacks on these datasets.
>
> > 3) In Sec 3.1, [...] Where do "3%" and "100 times" come from?
>
> Indeed, these numbers are not immediately clear from the figure, which provides the full picture by displaying TPR at every FPR. Table 1, however, reports the exact TPR number at ~3% FPR area for Waterbirds, showing that the vulnerability for the non-spurious group (0) is 0.22%, while for the spurious group (2) is 30.91%. We will clarify the reference of Table 1 from the section 3.1.
>
> > 4) The claim of "spurious groups can be up to 100 times more vulnerable to privacy attacks than non-spurious groups" [...] is exaggeration and should not include in the abstract.
>
> We will rephrase it as "spurious groups can be more vulnerable to privacy attacks than non-spurious groups".
>
>
>
> [1] Bengio, Yoshua, Ian Goodfellow, and Aaron Courville. Deep learning. MIT press, 2017
>
> [2] Prince, Simon JD. Understanding deep learning. MIT press, 2023
>
> [3] Yang, Yuzhe, et al. Change is Hard: A Closer Look at Subpopulation Shift. International Conference on Machine Learning, 2023"

---

> > ### Comment · Reviewer_9gk7 · 2024-11-25
> >
> > Thanks for the authors' reply. However, I still have several concerns.
> >
> > 1. Hyperparameter Selection.
> > The authors provide explanations for their hyperparameter choices. However, the varying hyperparameter settings across different experiments undermine the validity of Finding V, as it lacks a standardized comparison framework. Ref [1] and [2] given in the authors' comments are about how to choose hyperparameters, and these citations do not address the need for consistent experimental conditions to ensure fair comparisons.
> >
> > 2. New content. The authors updated one experiment and some related work.
> > However, the contents are not well structured, and the current version of the paper does not have enough proof to show that each finding is convincing and well supported.
> >
> > Based on the above concerns, I will keep the original score.

---

> > > ### Author Response · Authors · 2024-11-25
> > >
> > > > Hyperparameter Selection. The authors provide explanations for their hyperparameter choices. However, the varying hyperparameter settings across different experiments undermine the validity of Finding V, as it lacks a standardized comparison framework.
> > >
> > > For each model, we evaluate 12 different hyperparameter settings (learning rate: [1e-1, 1e-2, 1e-3, 1e-4]; weight decay: [1e-1, 1e-2, 1e-3]) and select the best configuration. We believe this approach ensures a robust and fair comparison, in particular compared to the prior works. If the reviewer has specific sources regarding the __standardized comparison framework__ that should be followed, we would appreciate it if they could share them.
> > >
> > > > [...] Contents are not well structured, and the current version of the paper does not have enough proof to show that each finding is convincing and well supported.
> > >
> > > We kindly request the reviewer to clarify which parts of the paper they find to be not well-structured and which findings they feel are insufficiently supported. Specific examples or detailed feedback can help us address these concerns.

---

### Official Review · Reviewer_E78d · 2024-10-22

**Soundness:** 2
**Presentation:** 3
**Contribution:** 2
**Rating:** 5
**Confidence:** 4

**Summary:**

The authors uncover that spurious groups in datasets (i.e., groups affected by spurious correlations) are significantly more vulnerable to membership inference attacks (MIA) than non-spurious groups. Meanwhile, as the task complexity decreases (e.g., fewer target classes in the dataset), the privacy leakage for spurious groups remains constant or worsens, while leakage for other groups reduces. Moreover, despite improvements in group performance disparity through methods like DRO, DFR, and DP, these methods fail to address the privacy disparity among spurious groups. They also show that architectures like Vision Transformers (ViTs) are not necessarily more robust against spurious correlations than convolutional models under fair experimental setups.

**Strengths:**

- The paper focuses on the connection between spurious correlations and privacy leakage, an underdeveloped topic in trustworthy machine learning.
- The paper presents several interesting observations.
- The paper is well-organized and easy to follow.

**Weaknesses:**

1. The motivation behind evaluating privacy disparities among subgroups (spurious vs. non-spurious groups) is unclear. While the paper shows that existing methods (DRO, DFR, or DP-SGD) may not fully address these privacy gaps, it's unclear why privacy parity across subgroups could be a priority. Why should we care about these gaps?

2. While the authors present technically interesting observations about privacy disparities, the results are more like experimental reports. It’s unclear how these findings can contribute to the field. For instance, could they inspire new defense strategies? Or inform advanced attack methods?

3. The conclusions in Sections 3 to 5 are important but not well-supported as they rely on limited experiments, datasets, and methods, which is not convincing. For example, Section 4.2 states that DP fails to protect certain vulnerable groups in the data. The results contradict previous research suggesting that DP can protect sample vulnerabilities by preventing memorization. The finding comes from a single experimental setup. The scope is too narrow to fully support such a broad conclusion. More ablation studies are needed.

4. The paper needs more detailed discussions and comparisons to better support the conclusions. It’s unclear how these findings are connected to current studies. For example, in Section 3, there is related work on subgroup evaluations of model fairness and privacy with MIA—are the findings consistent? For Section 4, previous studies have explored the utility tradeoffs of DP methods—do they also show a failure to protect samples? Similarly, in Section 5, prior research compares the model robustness of Vision Transformers and CNNs—do those results align? More discussion is needed across these sections to connect the findings with existing work.

5. The related work section feels too limited, given the paper covers multiple topics.

**Questions:**

see above

---

> ### Author Response · Authors · 2024-11-17
>
> We thank the reviewer for the insightful comments.
>
> > 1) [...] it's unclear why privacy parity across subgroups could be a priority. Why should we care about these gaps?
>
> Prior to our work, it was unclear what are the privacy implications of spurious correlations in natural data. We find that spurious correlation cause leads to spurious privacy leakage, which raises additional security and ethical challenges for sensitive domains. For example, after a privacy audit with a "low overall privacy risk" result, we may naively conclude that our model satisfies the privacy requirements. However, the privacy disparity (or variance) between subpopulations can be high, which breaks the privacy requirements when analyzing per-subpopulation risks (e.g. GDPR emphasizes fairness and equal treatment). Overlooking these gaps means assuming all the data have the same importance (which is not true for real-world data [1]). Therefore, we argue that we should care about these gaps to precisely understand the risks our models carry, to encourage the research of more robust defenses, and to improve the auditing process. We will revise our introduction to clarify the motivations.
>
> > 2) While the authors present technically interesting observations about privacy disparities [...] It’s unclear how these findings can contribute to the field. For instance, could they inspire new defense strategies? Or inform advanced attack methods?
>
> We will expand the discussion section on how our findings can contribute to the field:
>
> - We show that spurious correlations represent a privacy concern for real-world data, highlighting the issue and the importance of privacy disparity (see above Q1).
> - We are the first to show that robust methods (e.g. DRO, DFR) mitigate spurious correlations but not spurious privacy leakage. This result highlights the limitations of current defenses (e.g. spurious robust methods and also DP-SGD) from the privacy perspective, encouraging the community to develop better solutions.
> - Given the previous contributions, we suggest a potential future work direction that exploits spurious correlation for a stronger attack. For example, it may be possible to exploit spurious correlation to carry a data poisoning attack: adding poisoned samples to enhance the spurious correlation of a specific target subgroup.
>
> > 4) The paper needs more detailed discussions and comparisons to better support the conclusions. It’s unclear how these findings are connected to current studies. For example, in Section 3, there is related work on subgroup evaluations of model fairness and privacy with MIA—are the findings consistent? For Section 4, previous studies have explored the utility tradeoffs of DP methods—do they also show a failure to protect samples? Similarly, in Section 5, prior research compares the model robustness of Vision Transformers and CNNs—do those results align? [...]
>
> We thank the reviewer for suggesting how to expand our discussion and related works section.
>
> For section 3, we show the presence of privacy disparity in real-world spurious correlated data, connecting our results with prior research focused on privacy and fairness [2, 3, 4], where they also found the presence of privacy disparity between subpopulations within the fairness models.
>
> For section 4, we add the following paragraph: Although robust methods successfully mitigate spurious correlations and can mildly reduce the overall privacy at low FPR ("T" rows), they do not affect the privacy disparity. The leakage for the spurious groups is consistently at a similar level for all three training methods across datasets. These results complement the findings from [4], whose analysis is limited to overall privacy.
>
> For section 5, the main point exactly discusses (or revisits) the past work from Ghosal & Li 2024. Contrary to their conclusions, we find that ViTs do not outperform CNNs in mitigating spurious correlations, highlighting the importance of fair benchmarking.
>
> > 3) The conclusions in Sections 3 to 5 are important but not well-supported as they rely on limited experiments [...]
>
> See overall comment 2).
>
> > 5) The related work section feels too limited, given the paper covers multiple topics.
>
> See overall comment 1).
>
>
> [1] Feldman, Vitaly, and Chiyuan Zhang. What neural networks memorize and why: Discovering the long tail via influence estimation. Advances in Neural Information Processing Systems, 2020
>
> [2] Bogdan Kulynych, et al. Disparate vulnerability to membership inference attacks. Proceedings on Privacy Enhancing Technologies, 2022
>
> [3] Da Zhong, et al. Understanding disparate effects of membership inference attacks and their countermeasures. In Asia Conference on Computer and Communications Security, 2022
>
> [4] Huan Tian, et al. When fairness meets privacy: Exploring privacy threats in fair binary classifiers via membership inference attacks. In International Joint Conference on Artificial Intelligence, 2024

---

> > ### Comment · Reviewer_E78d · 2024-11-21
> >
> > Thank you for the reply. However, currently, I still would like to maintain my score for the following concerns:
> >
> > **Motivation and contribution**: Privacy disparities across subgroups are significant issues. However, they are not entirely unexplored. Existing studies have examined the intersection of privacy and fairness, as highlighted in [1]. Specifically, Chang [2] was the first to evaluate subgroup-specific privacy vulnerabilities using MIA, demonstrating that minority subgroups experience higher privacy leakage. **The manuscript should acknowledge prior contributions and clearly differentiate their findings (Finding 1) from Chang's work**.
> >
> > From the revised manuscript, I understand that, in this paper, authors observed that "The leakage for the spurious groups is consistently at a similar level for all three training methods (DRO, DFR, DP) across datasets." However, as mentioned in the initial comments, why should we be concerned about these similar gaps? Prior studies have already told us about privacy disparities across subgroups, and we can mitigate these privacy leakages by considering minority subgroups.
> >
> >
> > **Experiments**: The authors claimed several observations. However, some of the findings are similar to previous studies. Others are not well supported and not convincing enough due to the limited experiment settings. For example,
> > - **Finding 1**, Chang [2] already demonstrated that minority subgroups are more vulnerable to privacy leakage.
> > - **Finding 2**: Using the number of classes as a proxy for task complexity is too simple. As mentioned in [3], data privacy leakage (MIA) is closely connected to data outliers due to model memorization. Other factors can challenge their conclusions. For instance, what about simple datasets with few outliers vs datasets with more classes but more outliers? For per-sample privacy evaluations, the privacy leakage is primarily determined by whether the sample itself is an outlier, rather than the number of classes in the dataset.
> > - **Findings 3 and 4**, as mentioned earlier, authors should present more ablations studies to support their claims.
> > - **Finding 5** revisits Ghosal and Li’s work, and **Finding 6** is based solely on experiments with one dataset, Waterbirds, which is too limited.
> >
> > I recommend the authors focus on presenting their core findings rather than overselling their stories/contributions.
> >
> > [1] Duddu, Vasisht, Sebastian Szyller, and N. Asokan. "Sok: Unintended interactions among machine learning defenses and risks."2024 IEEE Symposium on Security and Privacy (SP). IEEE, 2024.
> >
> > [2] Chang, Hongyan, and Reza Shokri. "On the privacy risks of algorithmic fairness."2021 IEEE European Symposium on Security and Privacy (EuroS&P). IEEE, 2021.
> >
> > [3] Carlini, Nicholas, et al. "The privacy onion effect: Memorization is relative."Advances in Neural Information Processing Systems 35 (2022): 13263-13276.

---

> > > ### Author Response · Authors · 2024-11-22
> > >
> > > Thank you again for the reply.
> > >
> > > > Motivation and contribution: Privacy disparities across subgroups are significant issues. However, they are not entirely unexplored. [...] Specifically, Chang [2] was the first to evaluate subgroup-specific privacy vulnerabilities using MIA, demonstrating that minority subgroups experience higher privacy leakage. The manuscript should acknowledge prior contributions and clearly differentiate their findings (Finding 1) from Chang's work.
> > >
> > > Our related work section already connects our results with other studies on fairness and privacy. However, we want to emphasize that studying spurious correlations is related to, but distinct from, fairness. From a causal graph perspective, spurious correlations aim to disentangle causal and non-causal variables, while fairness often targets sensitive attributes with societal implications, making it a subset of spurious correlations. We appreciate the reviewer’s suggestion and will add the following discussion with Chang's work:
> > >
> > > Chang et al. (2020) were the first to evaluate the intersection of fairness and privacy, observing that minority subgroups exhibit higher privacy leakage. Our results extend these findings by showing that spurious groups are more susceptible to MIA in natural, larger-scale datasets. Additionally, we demonstrate that spurious mitigation methods (DRO, DFR) do not affect privacy leakage at low FPR. This result contradicts Chang’s claim that enforcing fairness increases leakage and prompts a re-evaluation of their findings, as they relied on suboptimal evaluation metrics: aggregated metrics combined with threshold-based MIA.
> > >
> > > > [...] "The leakage for the spurious groups is consistently at a similar level for all three training methods (DRO, DFR, DP) across datasets." However, as mentioned in the initial comments, why should we be concerned about these similar gaps? [...]
> > >
> > > It is a reasonable assumption that suppressing spurious correlations might also help to address privacy disparities. However, our results show that it is not true. While DRO and DFR successfully mitigate spurious correlations (i.e. improving WGA), they do not affect spurious privacy leakage. This suggests that spurious correlation mitigation and privacy protection require better approaches, highlighting a gap in current methodologies.
> > >
> > > > Finding 1, Chang already demonstrated that minority subgroups are more vulnerable to privacy leakage.
> > >
> > > See the above discussion regarding Chang's work.
> > >
> > > > Finding 2: Using the number of classes as a proxy for task complexity is too simple. [...] For instance, what about simple datasets with few outliers vs datasets with more classes but more outliers? For per-sample privacy evaluations, the privacy leakage is primarily determined by whether the sample itself is an outlier, rather than the number of classes in the dataset.
> > >
> > > We agree that the degree of "outlier" influences privacy leakage, but it is not the only factor. For example, why does CIFAR100 exhibit significantly higher privacy leakage than CIFAR10 (almost an order of magnitude difference at low FPR [1])? In our Finding 2, we did not alter the dataset size or introduce outliers. Instead, we fixed the dataset and controlled only the number of classes to isolate its effect under spurious correlation. Our results show increased spurious privacy leakage as the number of classes decreases, which we attribute to the influence of spurious correlations (see Figure 3).
> > >
> > > > Findings 3 and 4, as mentioned earlier, authors should present more ablations studies to support their claims.
> > >
> > > For Finding 4, we have extended the experiments by adding two new datasets (see our overall response). For Finding 3, we already include 12 distinct experimental setups, along with the additional memorization analysis in Figure 4. If there are specific ablations the reviewer would like us to conduct, we would be happy to consider them.
> > >
> > > > Finding 5 revisits Ghosal and Li’s work, and Finding 6 is based solely on experiments with one dataset, Waterbirds, which is too limited.
> > >
> > > Both Findings 5 and 6 serve to provide counterexamples to prior work. Specifically, for Finding 6, previous studies suggested that the strongest attack occurs when the shadow model matches the target architecture (evaluated only on CIFAR10 [1]). We identify a sufficient counterexample in the spurious-correlated Waterbirds dataset. While this is based on one dataset, one counterexample is sufficient to challenge the generality of prior works.
> > >
> > > We hope our clarifications address your concerns and can help you to reassess our work.
> > >
> > > [1] Nicholas Carlini et al. Membership inference attacks from first principles. In Symposium on Security and Privacy, 2022.
> > > [2] Yiyong Liu et al. Membership inference attacks by exploiting loss trajectory. In Conference on Computer and Communications Security, 2022a.

---

### Official Review · Reviewer_Kin4 · 2024-10-30

**Soundness:** 3
**Presentation:** 4
**Contribution:** 3
**Rating:** 8
**Confidence:** 5

**Summary:**

This paper shows experimentally that in data with spurious correlations and imbalanced groups, the minority groups are more susceptible to membership inference attacks than the majority groups. This high level message -- that out-of-distribution examples tend to be less private -- was known before: see for example, Figure 13 in the LiRA paper, [1], where this fact is exploited to design better privacy tests, as well as [2]. However what I like about this paper is that they do a very comprehensive experimental study on real data, and show a number of additional conclusions, and hence there is value in accepting it.
[1] https://arxiv.org/abs/2210.02912
[2] https://arxiv.org/abs/2202.05189

**Strengths:**

1. The paper presents highly comprehensive experiments that are quite well done. The results are well-presented, and well-supported by experimental evidence.

**Weaknesses:**

1. I would urge the authors to make some of the details a little bit more transparent in the main body. One difference between the standard setting and this work is that MIAs are used for fine-tuning and not pre-training data. This may mean that the fine-tuned datasets are very small per model. One of the best-kept secrets in LIRA-style membership inference is that the MIA is always carried on models that are trained on only a subset of the data, and making that subset bigger leads to worse "privacy loss".

Since this paper is further using MIA on models fine-tuned on a small amount of data, what is the size of the data that the model is fine-tuned on?

2. I am also not sure how meaningful the differential privacy results are -- since here epsilon=128. That kind of value for epsilon offers quite negligible privacy. That being said the remaining results are quite interesting.

**Questions:**

See Weakness 1.

---

> ### Author Response · Authors · 2024-11-17
>
> We thank the reviewer for the support and for the insightful review.
>
> > 1) I would urge the authors to make some of the details a little bit more transparent [...] difference between the standard setting and this work is that MIAs are used for fine-tuning and not pre-training data [...]
>
> We improve the clarity in section 3.1 as follows:
>
> Experiment setup. [...] We use the pretrained ResNet50 (He et al., 2016) on ImageNet1k from the timm library and finetune the spurious datasets with random crop and horizontal flip augmentations. Our setting differs from the standard settings (i.e. training from random initialization) by initializing our models pretrained on public data to facilitate the optimization process.
>
> > 2) [...] what is the size of the data that the model is fine-tuned on?
>
> The column _Group (n)_ from Table 1 reports the size of data for each group and total (T). The total is used for finetuning and represents approximately 50% of the original training data (as in LiRA). We noticed and corrected the values for the dataset MultiNLI, which was reporting 100% of the data.
>
> > 3) I am also not sure how meaningful the differential privacy results are -- since here epsilon=128. That kind of value for epsilon offers quite negligible privacy. That being said the remaining results are quite interesting.
>
> Indeed, a high epsilon {128} may offer negligible privacy. We included these results for completeness, showing how they compare to low privacy budget settings. We invite the reviewer to check also our overall comment 2) where we strengthen our differential privacy results with additional experiments.

---

> ### Comment · Reviewer_Kin4 · 2024-11-22
>
> Thank you for the clarification and the new experiments. Even though some of the findings were there piecemeal in previous work, I still do think that there is value in this paper as it carries out a comprehensive set of experiments on this topic. I will retain my score.

---

### Official Review · Reviewer_7w3C · 2024-11-04

**Soundness:** 2
**Presentation:** 3
**Contribution:** 2
**Rating:** 5
**Confidence:** 4

**Summary:**

This paper investigates the finding that groups influenced by spurious correlations in datasets are more vulnerable to membership inference attacks (MIA) than other groups. The study also shows that Vision Transformers (ViTs) are not necessarily better than convolutional models at handling spurious correlations. The paper highlights how bias in neural networks affects privacy risks.

**Strengths:**

1. The paper attempts to undermine the relationship between bias an d privacy leakage. The key findings are well articulated via examperiments.
2. The paper is overall well written and easy to follow.

**Weaknesses:**

1. The motivation is not very clear. What is the reason to assess privacy disparities between spurious and non-spurious subgroups?
2. The related work section is too brief considering the multiple topics the paper covers. Expanding it to include more of the relevant literature would strengthen the foundation. It is also unclear how the findings align with current research.
3. The experiments are not clearly explained. Including the choice of datasets and neural network models.

**Questions:**

Refer to the weakness section.

---

> ### Author Response · Authors · 2024-11-17
>
> We thank the reviewer for the feedback.
>
> > 1) [...] What is the reason to assess privacy disparities between spurious and non-spurious subgroups?
>
> Prior to our work, it was unclear what are the privacy implications of spurious correlations in natural data. We find that spurious correlation cause leads to spurious privacy leakage, which raises additional security and ethical challenges for sensitive domains. For example, after a privacy audit with a "low overall privacy risk" result, we may naively conclude that our model satisfies the privacy requirements. However, the privacy disparity (or variance) between subpopulations can be high, which breaks the privacy requirements when analyzing per-subpopulation risks (e.g. GDPR emphasizes fairness and equal treatment). Overlooking these gaps means assuming all the data have the same importance (which is not true for real-world data [1]). Therefore, we argue that we should care about these gaps to precisely understand the risks our models carry, to encourage the research of more robust defenses, and to improve the auditing process. We will revise our introduction to clarify the motivations.
>
> > 2) The related work section is too brief considering the multiple topics the paper covers. [...] It is also unclear how the findings align with current research [...]
>
> See overall comment 1) where we expand the related work section.
>
> Then, we expand our discussion in each section as follows:
>
> For section 3, we show the presence of privacy disparity in real-world spurious correlated data, connecting our results with prior research focused on privacy and fairness [2, 3, 4], where they also found the presence of privacy disparity between subpopulations within the fairness models.
>
> For section 4, we add the following paragraph: Although robust methods successfully mitigate spurious correlations and can mildly reduce the overall privacy at low FPR ("T" rows), they do not affect the privacy disparity. The leakage for the spurious groups is consistently at a similar level for all three training methods across datasets. These results complement the findings from [4], whose analysis is limited to overall privacy.
>
> For section 5, the main point exactly discusses (or revisits) the past work from Ghosal & Li 2024. Contrary to their conclusions, we find that ViTs do not outperform CNNs in mitigating spurious correlations, highlighting the importance of fair benchmarking.
>
> > 4) The experiments are not clearly explained. Including the choice of datasets and neural network models
>
> We clarify the section 3.1 by explaining the choice of datasets:
>
> Experiment setup. We choose the datasets that are commonly used by the spurious correlation community (see Table 2 of [1]): Waterbirds, CelebA, FMoW, and MultiNLI. These datasets contain real-world spurious correlations, diverse modalities, and different target complexity. Moreover, to the best of our knowledge, we are the first to study MIA on these datasets.
>
> Next, we clarify our choice of models. All of our experiments related to images include ResNet50 as it is a solid baseline across the literature for both MIA and spurious correlation [1]. The BERT model is used for the text dataset MultiNLI and is also commonly used as the baseline in spurious correlation works [1]. Lastly, for Section 5, we compare eight different neural networks. Contrary to previous works comparing similar models (e.g. ResNet, DenseNet, MobileNet...) [2, 3], we include only models that are sufficiently "diverse". For example, we compare models from different families (convolution and transformers), with different pretraining (supervised and self-supervised), and released at different time periods (e.g. ResNet and its successor ConvNext).
>
> In case the reviewer still has doubts, may you help us to pinpoint exactly which part requires additional clarification?
>
>
> [1] Yang, Yuzhe, et al. Change is Hard: A Closer Look at Subpopulation Shift. International Conference on Machine Learning, 2023
>
> [2] Yiyong Liu et al. Membership inference attacks by exploiting loss trajectory. In Conference on Computer and Communications Security, 2022a.
>
> [3] Nicholas Carlini et al. Membership inference attacks from first principles. In Symposium on Security and Privacy, 2022

---

### Author Response · Authors · 2024-11-17
**Overall Comment by Authors**

The following paragraphs collect the answers to a few common doubts raised by different reviewers.

__1) Expanded literature__

Our literature study is focused on 3 different topics: 1) membership inference, 2) spurious correlation, 3) papers with a higher-level perspective, related to privacy and safety ML. For pedagogical reasons, we decided to divide these three topics into three different (sub)sections, as close as possible to the relevant part of the paper. The topics 1 and 2 are in the background section as they provide the notions for understanding the work, while topic 3 is discussed in the related work section. Together, these sections currently include ~25 citations. Further, in response to the feedback, we will expand the sections with additional related works that enhance the context and additional discussion with prior works. Lastly, if the reviewers have any specific work that we should have cited in mind, we will follow your suggestions and add them to the right spot in our paper.

__2) Additional results__

We add new experiments for the section 4.2 focused on differential privacy:

We train DP-SGD target models on CelebA and FMoW for 50 epochs using batch size 1024 and grid searching the lr in [1e-1, 1e-2], \epsilon in [1, 2, 8, 32, 128], and \delta of 1e-5. We reuse the shadow models trained in section 3.1 for LiRA. Consistent with our findings for Waterbirds, low ϵ (=1, 2) hinders the generalization, while high ϵ better mitigates spurious correlations. However, in terms of privacy vulnerability, all models across different privacy budget levels exhibit similar levels of vulnerability, highlighting the challenges of applying DP while retaining the utility of spurious data.

Table 1. Models trained on the FMoW dataset fail to generalize under a low privacy budget and are therefore ignored. The privacy vulnerability remains similar across different levels.
| eps | delta  | test acc (avg)  | test acc (worst) | tpr\@0.1%fpr (avg) | tpr\@0.1%fpr (worst) |
|-----|--------|-------|-----------|------|------|
| 1   | 1e-5   | 13.01 | 9.45      | - | - |
| 2   | 1e-5   | 19.58 | 12.53     | - | - |
| 8   | 1e-5   | 38.64 | 26.34     | 6.66 | 6.47 |
| 32  | 1e-5   | 43.58 | 27.77     | 6.83 | 6.60 |
| 128 | 1e-5   | 43.95 | 29.73     | 6.82 | 6.60 |

Table 2. Models trained CelebA dataset are more spurious robust as the privacy budget is relaxed. However, the privacy level remains similar across budgets.
| eps | delta  | test acc (avg)  | test acc (worst) | tpr\@0.1%fpr (avg) | tpr\@0.1%fpr (worst) |
|-----|--------|--------|-----------|------|------|
| 1   | 1e-5   | 91.45  | 25.56     | 0.77 | 4.38 |
| 2   | 1e-5   | 92.79  | 40.00     | 0.75 | 4.38 |
| 8   | 1e-5   | 95.46  | 35.56     | 0.75 | 4.22 |
| 32  | 1e-5   | 95.64  | 41.11     | 0.77 | 4.06 |
| 128 | 1e-5   | 95.54  | 47.22     | 0.76 | 4.38 |

For the other sections, we argue that our experiments sufficiently support the findings. For section 3.1, our MIA setting includes four datasets with real-world spurious correlations, different target complexity, and diverse modalities. For section 4.1, the setting compares three different training methods on four datasets (12 different experiment setups). For section 4.2, we added the new results as described above. Lastly, for section 5, we fairly compare eight different model architectures, finding one sufficient counterexample to warrant revisiting the claim from Ghosal & Li (2024).

---

### Author Response · Authors · 2024-11-19
**Rebuttal Revision**

First of all, many thanks to all the reviewers for your valuable feedback. We have improved our manuscript with the following changes:

- Add two new experiments for section 4.2
- Expand the related work section
- Expand the discussions with past research in sections 3, 4, 5
- Motivate the importance of our research
- Clarify the choice of datasets and models
- Clarify the results in section 3.1

---

### Author Response · Authors · 2024-12-04

We provide a summary of this rebuttal period.

Reviewers find the topic of "spurious privacy leakage" interesting (9gk7, E78d, Kin4). In particular, reviewer Kin4 states that our results are well-presented and well-supported by experimental evidence. Reviewer E78d also finds our results interesting, and reviewer 7w3C notes that they are well-articulated via experiments. Moreover, all reviewers praised the presentation style. During the rebuttal phase, we addressed all concerns raised by the reviewers which can be summarized into three main points:

- __More experiments are needed (9gk7, E78d).__
We believe that the quality of datasets matters more than their quantity. For example, experiments based on CIFAR10/100 and ImageNet are expected to yield similar results due to their similarity. In contrast, our experiments are based on diverse real-world datasets, covering different tasks, modalities (vision and text), and target complexity (2, 3, 62 classes). Consistent observations across these diverse datasets provide even stronger evidence to support our claims. Additionally, during the rebuttal, we added 2 more experiments to support finding 4 (as requested by E78d).

- __More literature discussion is needed (9gk7, E78d, 7w3C)__.
For pedagogical reasons, we note that some aspects of related work are included in the background section. Additionally, we expanded the related work section and incorporated discussions of prior studies into the results of each section.

- __Clarification on the motivation (E78d, 7w3C)__.
We clarified the motivation of our work in the introduction.

Many thanks to everyone for your service!

---

### Meta-Review · Area_Chair_4iBk · 2024-12-21

**Metareview:**

The authors investigate the impact of spurious correlation bias on privacy vulnerability. They introduce a phenomenon called, spurious privacy leakage, and report that spurious groups can be more vulnerable to privacy attacks than non-spurious groups.

Reason for decision: I found this paper's main message a repetition of a relatively well known phenomenon: out-of-distribution examples tend to be less private (see examples given by Reviewer Kin4). What is valuable is that the authors provided comprehensive experimental study for this message. However, perhaps this is not such a strong or surprising message to most of the privacy/ML audiences.

**Additional Comments On Reviewer Discussion:**

Reviewer 7w3C mentioned the related work section is too thin. The authors rebutted by expanding the section, while the reviewer has not responded to the rebuttal.

After some conversations between the authors and Reviewer 9gk7, the reviewer still thinks this paper needs better structure and organization.

Probably this paper will be better shaped if they incorporated all of these feedbacks from the reviewers for their next submission to another ML venue.

---

### Decision · Program_Chairs · 2025-01-22

Reject